# Foxi2 and Sox3 are master transcription regulators that control ectoderm germ layer specification in *Xenopus*

Clark L. Hendrickson[1], Ira L. Blitz[1], Amina Hussein[1,2], Kitt D. Paraiso[1,2], Jin S. Cho[1], Michael W. Klymkowsky[3], Matthew J. Kofron[4,5], Ken W. Y. Cho[1,2]*

**1** Developmental and Cell Biology, University of California, Irvine, California, United States of America, **2** Center for Complex Biological Systems, University of California, Irvine, California, United States of America, **3** Molecular, Cellular, and Developmental Biology, University of Colorado, Boulder, Colorado, United States of America, **4** Department of Pediatrics, University of Cincinnati College of Medicine, Cincinnati, Ohio, United States of America, **5** Division of Developmental Biology, Cincinnati Children's Hospital Medical Center, Cincinnati, Ohio, United States of America

* kwcho@uci.edu

## Abstract

Germ layer specification represents a critical transition where pluripotent cells acquire lineage-specific identities. We identify the maternal transcription factors Foxi2 and Sox3 to be pivotal master regulators of ectodermal germ layer specification in *Xenopus*. Ectopic co-expression of Foxi2 and Sox3 in prospective endodermal tissue induces the expression of ectodermal markers while suppressing mesendodermal markers. Transcriptomic analyses reveal that Foxi2 and Sox3 jointly and independently regulate hundreds of ectodermal target genes. During early cleavage stages, Foxi2 and Sox3 pre-bind to key *cis*-regulatory modules (CRMs), marking sites that later recruit Ep300 and facilitate H3K27ac deposition, thereby shaping the epigenetic landscape of the ectodermal genome. These CRMs are highly enriched within ectoderm-specific super-enhancers (SEs). Our findings highlight the pivotal role of ectodermal SE-associated CRMs in precise and robust ectodermal gene activation, establishing Foxi2 and Sox3 as central architects of ectodermal lineage specification.

## Introduction

A major breakthrough in biology was the first animal (*Xenopus laevis*) cloning experiment, which proved that differentiated cells retain the full genetic potential to develop into an entire organism [1]. This result demonstrated the concept of genomic equivalence, where cellular differentiation is governed by gene regulation rather than by irreversible modifications to the genome, such as differential loss of subsets of genes in cell lineages. These experiments further revealed that the egg cytoplasm contains necessary maternal regulators to reprogram the epigenetic memory of the

**Data availability statement:** GEO accession numbers for data generated in this paper: GSE288636 (RNA-seq), GSE288637 (ChIP-seq), and GSE288638 (snRNA-seq).

**Funding:** This research was funded by National Institutes of Health (GM139617 and HD109696 to K.W.Y.C.) and National Science Foundation (1755214 to K.W.Y.C.). Additional support was provided by a GAANN Fellowship from the U.S. Department of Education (P200A220015 to C.L.H.). The funders had no role in study design, data collection and analysis, decision to publish, or preparation of the manuscript.

**Competing interests:** The authors have declared that no competing interests exist.

**Abbreviations:** ChIP, chromatin immuno-precipitation; COV, coefficient of variation; CRM, *cis*-regulatory module; FSE, Foxi2, Sox3, and Ep300; GO, gene ontology; MOs, morpholino antisense oligonucleotides; REs, regular enhancers; ROSE, rank-ordered super enhancer; SEs, super-enhancers; TFs, transcription factors; UMAP, Uniform Manifold Approximation Projection; ZGA, zygotic genome activation; RT, reverse transcription.

differentiated cell nuclei to a totipotent state. Subsequently, it was shown in mammalian cells that networks of master regulator transcription factors (TFs), including most notably Klf4, Myc, Pou5f1, and Sox2, are critical for embryonic pluripotency [2]. This highlights the essential role of multiple TFs working in concert to orchestrate cellular identity, lineage specification, and reprogramming.

Additionally, discovery of these pluripotency TFs among many permutations highlights that not all TFs hold equal importance in development. TFs are considered master regulators when they play a more critical role in regulating specific differentiation pathways because they occupy a higher position within a gene regulatory program controlling downstream events in differentiation and maintaining pluripotency. One of the earliest TFs to support the concept of master regulators was Myod1, which was shown to drive fibroblasts into a muscle cell differentiation path [3]. Other examples include Spi1 (Pu.1), which specifies myeloid and B cell lineages in hematopoiesis [4], Gata1 in erythroid lineage specification [5], and Pdx1 for pancreas organogenesis [6]. These master regulatory TFs play a pivotal role in determining the identity and function of specific cell types. Identifying master regulatory TFs in developmental biology is fundamentally important for comprehending the mechanisms of cell lineage segregation and advancing regenerative medicine to facilitate direct cell lineage conversion.

Our goal in this study is to determine the primary master regulators initiating ectodermal cell lineage differentiation at the highest positions in the programming of cell lineages. In vertebrates, germ layer specification is one of the earliest developmental decisions that pluripotent cells make. In amphibians, the three germ layers are organized along the animal–vegetal axis, where the ectoderm forms in the animal pole, the endoderm vegetally, and the mesoderm in the equator. The ectoderm comprises progenitor cell populations giving rise to the skin, nervous system, sensory organs, and numerous neural crest derivatives. Several experiments have demonstrated that *Xenopus* ectodermal cells of the blastula stage are pluripotent. Transplantation of blastula prospective ectodermal cells into the vegetal endoderm converts the ectodermal cells to endodermal fates [7]. In addition, explanted blastula ectodermal tissue ("animal caps") can be converted to mesendodermal cells *ex vivo* (organoids), by soaking them in activin, a TGFβ-superfamily ligand [8–10]. With specific combinations of growth factors and/or small molecules, blastula ectodermal explants can be programmed to differentiate into a wide range of cell types, including neural tissue, muscle cells, blood, heart, and endoderm primordia [11,12]. Although this ectodermal pluripotency persists through blastula stages, it is autonomously lost by early gastrula [13,14]. Therefore, ectodermal germ layer and sub-lineage specification is an ideal system to study the dynamic processes controlling cellular potency and differentiation.

Previous epigenetic studies have shown that the early embryonic genome lacks significant histone modifications before the blastula stage and therefore remains epigenetically naïve (reviewed in [15]). Histone modifications associated with active enhancers first appear during blastula stage, coinciding with major zygotic genome activation (ZGA) and the segregation of the pluripotent embryonic cells into

specialized ectoderm, mesoderm, and endodermal lineages. We hypothesize that maternal, locally expressed master TFs bind to specific *cis*-regulatory modules (CRMs) to activate or repress the transcription of the first wave of zygotic target genes, thus orchestrating the gene regulatory programs required for cell lineage specification.

Here, we reveal that the ectodermal gene regulatory program is orchestrated by maternally expressed, and ectodermally enriched master regulatory TFs, Foxi2 and Sox3. These TFs prebind to the CRMs of the naïve embryonic genome prior to ZGA, preceding the appearance of major histone modifications and transcription. This prebinding event coordinates the subsequent ectodermal cell differentiation program. The CRMs bound by Foxi2 and Sox3 are associated with ectodermal super-enhancers (SEs), which ensure robust target gene expression. Our findings are consistent with the view that Foxi2 and Sox3 function as master regulators of the ectodermal germ layer.

## Results

### Different epigenetic states accrue on ectoderm- and endoderm-expressed genes during ZGA

Epigenetic analyses of *Xenopus* blastula stage embryos show that the embryonic genome is largely devoid of the major histone tail modifications H3K27ac, H3K27me3, and H3K4me1 [16,17]. These marks begin to accrue during blastula stages, becoming widespread in the genome during gastrulation [17–20]. However, a limitation of whole-embryo epigenetic experiments, which homogenize the data from cells from all three germ layers, is the inability to determine whether these histone modifications occur in a spatially restricted manner.

To gain spatial information about histone modification states in different germ layers, early gastrulae (stage 10.5) were dissected into ectoderm and endoderm fragments, approximately 3 hours after the onset of ZGA (stage 8.5) (S1A and S1B Fig). Histone modifications H3K27ac and H3K27me3 were then examined around enhancers and promoters to understand their role in regulating the transcription of associated genes (S1C Fig). Using early embryonic time course [21], and early gastrula embryo dissection [22] RNA-seq data, we determined the top 250 zygotically expressed genes specifically enriched in either the ectoderm or endoderm. The germ layer-specific histone mark deposition of H3K27ac and H3K27me3 was plotted within a 25kb region upstream and downstream of the genes with associated peaks (Fig 1A). The mesoderm was excluded from the analysis due to contamination issues occurring across tissue boundaries.

Ectodermally expressed genes, including their up and downstream regions, are more highly decorated with the activating H3K27ac mark in the ectoderm compared to the endoderm (Figs 1A, 1B, and S1C). On the other hand, endodermally expressed genes are more highly marked by H3K27ac in the endoderm compared to the ectoderm. We also examined the distribution of the repressive mark H3K27me3 across both ectodermally and endodermally expressed genes. Endodermally expressed genes are strongly marked by H3K27me3 in ectoderm, but remain relatively unmarked by H3K27me3 in endoderm (Figs 1A, 1B and S1C). This H3K27me3 deposition suggests a mechanism of active repression of endodermally expressed genes in the ectoderm, which prevents their inappropriate expression outside their endodermal environment. In contrast, H3K27me3 marking of ectodermally expressed genes in both endoderm and ectoderm cells is low. Notably, Akker and colleagues [23] also observed H3K27me3 enrichment on a subset of lineage-enriched marker genes. However, germ layer-specific H3K27me3 modifications were not examined. Our findings, based on dissected tissues from early gastrula embryos, reveal tissue-specific accumulation of H3K27me3 at endodermal gene loci in the ectoderm, a pattern not mirrored for ectodermal genes in the endoderm.

Since H3K27ac marking of ectodermal and endodermal genes correlates well with their region-specific expression in developing embryos, we examined the binding of the histone acetyl transferase Ep300 (Figs 1C and S1D), which is a commonly used marker to identify functional enhancers [24]. In ectoderm explants, Ep300 binding is high in ectodermally expressed genes, while in endoderm explants it is preferentially enriched in endodermal gene binding (Fig 1C). Ep300 functions as a transcriptional co-activator, but it does not possess a DNA-binding domain. Therefore, the recruitment of Ep300 to specific genomic loci requires interactions with DNA-bound TFs. To identify potential Ep300 cofactors, TF motif enrichment search of Ep300-bound peaks was performed for both ectodermally

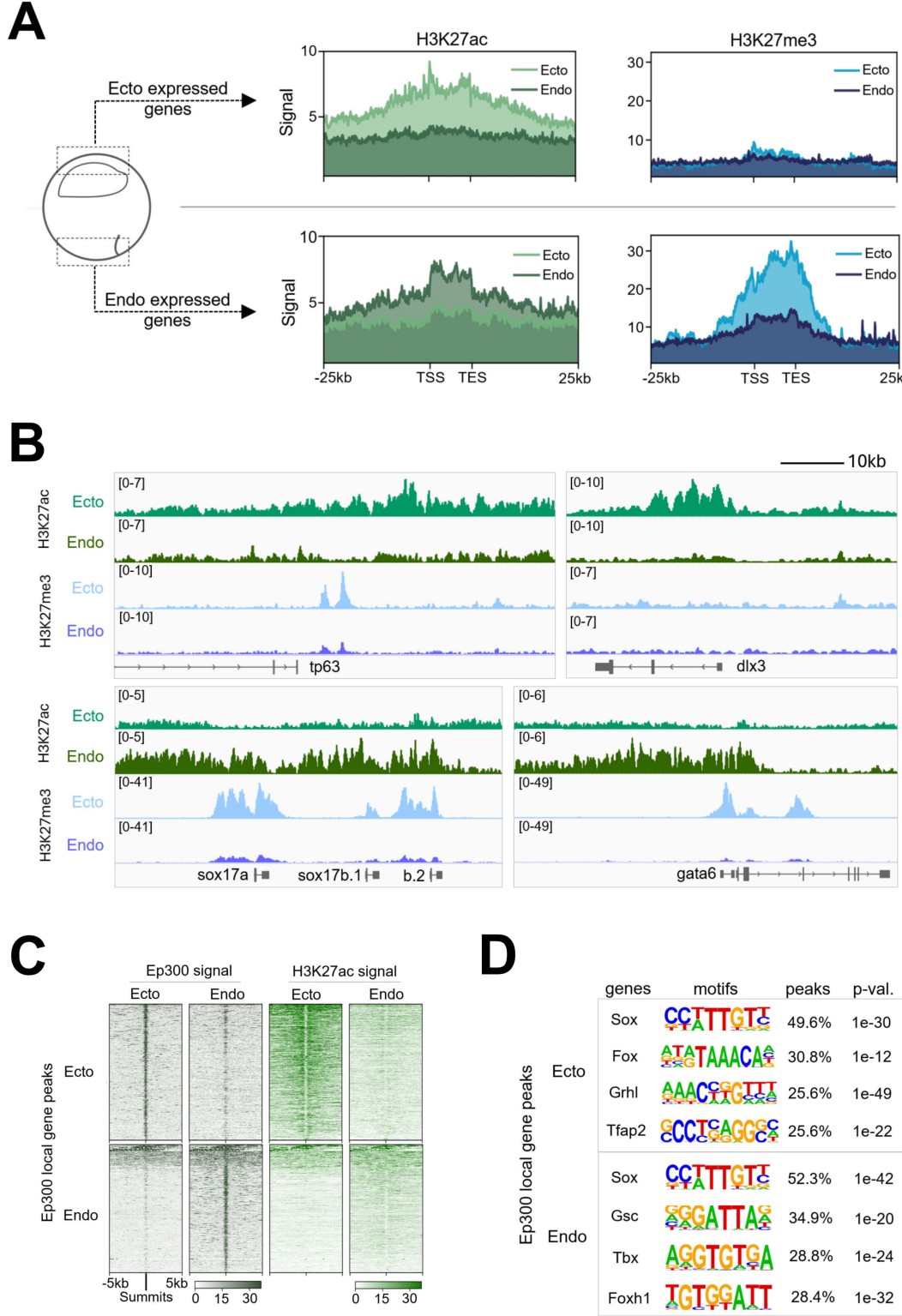

**Fig 1. Differential epigenetic regulation of ectodermal and endodermal genes in early gastrulae. (A)** Signal deposition of H3K27ac (left, green) and H3K27me3 (right, blue) across peaks associated with the top 250 zygotically expressed ectodermal (top) or endodermal (bottom) gene regions obtained from early gastrula ectoderm and endoderm dissections. **(B)** Genome browser tracks showing H3K27ac and H3K27me3 marks along

representative zygotically expressed ectodermal and endodermal genes. **(C)** Signal enrichment of Ep300 (left) and H3K27ac (right) across Ep300-bound regions associated with the top 250 zygotically expressed ectodermal (top) and endodermal (bottom) genes from early gastrula ectoderm and endoderm dissections. **(D)** Transcription factor motifs detected within Ep300 peaks within 20kb of top 250 ectodermal (top) and endodermal (bottom) genes. The data underlying this figure can be found in S1 Data and GSE288637.

and endodermally expressed gene regions. Sox, Fox, Grhl, and Tfap2 family motifs were most frequently identified in Ep300 peaks associated with ectodermally expressed genes, whereas the Sox, Gsc, Tbx, and Foxh1 motifs were most frequent in endodermally expressed genes (Fig 1D). These results suggest that gene families related to Sox and Fox TFs are critical in both ectodermal and endodermal development. Importantly, *foxi2* and *sox3* transcripts are highly expressed maternally (S2A Fig) and enriched in the ectoderm [20,22,25,26], implying their involvement, in not only regulating the ectodermal gene regulatory program, but also influencing epigenetic modifications of ectodermal genes.

### Foxi2 and Sox3 prebind future ectodermal CRMs before Ep300 recruitment

Previous work showed that maternal Foxi2 can bind to an upstream CRM of the *foxi1* gene to activate its expression [25]. We wished to further examine the genome-wide regulatory role of Foxi2 during early development. Chromatin immunoprecipitation (ChIP)-seq analysis of Foxi2 was performed in mid- to late blastula (st. 8, 9) and early gastrula (st. 10.5) embryos and identified a total of 16,427 bound regions over the entire time course (Fig 2A). There are 13,158 high-confidence Foxi2 bound regions in the mid blastula, 6,668 in the late blastula, and 6,039 in the early gastrula (Fig 2A). Overall, Foxi2 binding is dynamic, displaying unique (class I, III, VI), shared (class II, IV, V), as well as persistently occupied (class VII) binding patterns across stages (Fig 2A and 2B).

To characterize functionally relevant Foxi2 peaks, we examined the correlation of classified Foxi2 peaks with Ep300 and H3K27ac peaks in ectoderm and endoderm explants. Class IV, V, VI, and VII regions contain more H3K27ac deposition and Ep300 binding compared to classes I, II, and III (Figs 2B and S2B). In total, 3,050 Foxi2 peaks are Ep300 co-occupied, where class IV, V, VI, and VII Foxi2 peaks show the highest degree of overlap with both Ep300 and H3K27ac (Figs 2B, 2C and S2B). We bioinformatically assigned nearest zygotic genes to the Foxi2 peaks marked by Ep300, and examined whether these CRMs are linked to transcriptional activity of the genes. We find that class IV, V, VI, and VII genes become transcriptionally active during gastrulation, whereas class I, II, and III genes remain inactive (Fig 2D). These results suggest that Foxi2 peaks (1) are present at stage 10.5, either uniquely or overlapping with blastula stage binding, (2) are co-bound with ectodermal Ep300, (3) are decorated with ectodermal H3K27ac, and (4) are associated with genes expressed post ZGA, indicating that they represent active CRMs.

TFs form complexes on CRMs and typically act in combination. To predict potential co-factors that may work together with Foxi2, we characterized TF motifs enriched within Foxi2-bound regions. As expected, de novo motif analysis reveals Fox motifs are the most centrally enriched (Fig 2E). Besides the Fox motif, the next most abundantly present and centrally enriched motifs are Sox and Pou. These Fox/Sox/Pou motifs co-occur in 33% of Foxi2 peaks bound by Ep300, potentially suggesting complex formation (Fig 2F). Interestingly, while 48% of Foxi2 peaks contain only Fox/Sox motifs, just 7% contain only Fox/Pou motifs (Fig 2E and 2F).

Given Foxi2 and Sox3 high rate of motif co-occurrence, and their maternal expression, we assessed whether Foxi2 and Sox3 prebind functional ectoderm CRMs, before Ep300 recruitment and the onset of ZGA. Foxi2, Sox3, and Ep300 ChIP-qPCR analysis of 64-cell stage embryos (st. 6.5, ~3hpf) reveals that Foxi2 and Sox3, but not Ep300, occupy CRMs for *dlx5, foxi1, hgf, krt7, krt70, tfap2a, and tp63* (Fig 2G), which are expressed in the ectoderm of the early gastrula. This demonstrates that Foxi2 and Sox3 prebind to these ectodermal CRMs during cleavage stages, thus prior to ZGA (st. 8.5, ~4–4.5hpf), and before the recruitment of Ep300. This finding demonstrates that these maternal TFs initially recognize the CRMs of future ectodermally expressed genes during the cleavage stages.

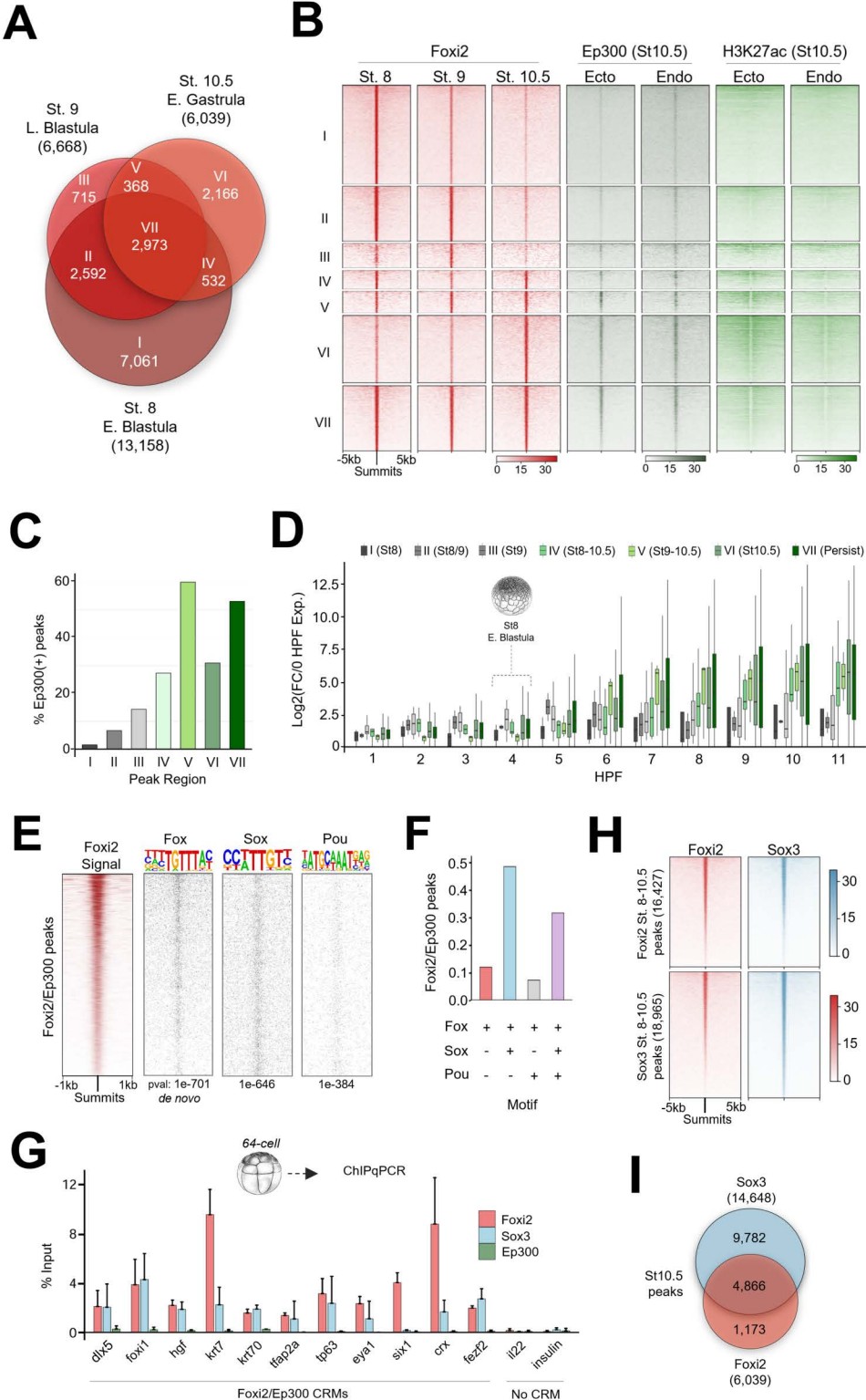

**Fig 2. Temporal dynamics and motif analysis of Foxi2 DNA binding and co-occupancy with Sox3 and Ep300 from blastula to early gastrula.**
**(A)** Venn diagram showing temporal dynamics of Foxi2 DNA binding, with ChIP-seq peaks categorized into Classes I–VII. **(B)** Analysis comparing Ep300 and H3K27ac signal from early gastrula ectoderm and endoderm dissections, within Foxi2 peak classification. **(C)** Proportion of Foxi2 class I–VII

peaks overlapping Ep300 peaks from early gastrula ectoderm. **(D)** Temporal RNA expression profiles of genes within 20 kb of Foxi2/Ep300 co-bound peaks in classes I–VII. **(E)** Top predicted Fox motif and best-matched Sox and Pou motifs identified within Foxi2/Ep300 peaks. **(F)** Proportion of Fox, Sox, and Pou motifs present in Foxi2/Ep300 peaks. **(G)** 64-cell embryo ChIP-qPCR analysis of Foxi2, Sox3, and Ep300 binding at early gastrula Foxi2/Ep300 co-bound regions. Error bars represent standard deviation from biological duplicates, ChIP samples were normalized by their percent recovery compared to input (non-pulldown) samples. **(H)** Total Foxi2 (top) and Sox3 (bottom) peaks from early blastula to early gastrula stages, with corresponding Foxi2 and Sox3 ChIP-seq signal. **(I)** Early gastrula Sox3 peak overlap with Foxi2 class IV–VII "active" peaks. The data underlying this figure can be found in S2 Data and GSE288637. *Xenopus* illustrations © Natalya Zahn [27].

## Active ectodermal CRMs are co-occupied by Foxi2 and Sox3

Given that 64-cell stage embryos (st. 6.5, ~3hpf) have not begun zygotic gene expression, we asked if early Foxi2 and Sox3 colocalization was maintained through ZGA. Sox3 ChIP-seq during the mid- through late blastula (st. 8–9) and the early gastrula (st. 10.5), yielded a total of 18,965 peaks, which are extensively co-occupied by Foxi2 (Figs 2H, S3A and S3B). Similarly, Sox3 is enriched across many of the 16,427 Foxi2-bound regions during this time window (Fig 2H). As development progresses from the mid-blastula to early gastrula, the binding of each TF increasingly bind to regions already occupied by the other TF (S3A Fig). We examined the dynamic binding of Foxi2 and Sox3 around several ecto-dermally expressed genes and observed that their co-occupancy is either maintained or increases from the early blastula to the early gastrula (S3B Fig). By the early gastrula stage, Sox3 is enriched at approximately 81% of Foxi2 class IV, V, VI, VII peaks (Fig 2I), which show strong overlap with Ep300 binding and are associated with robust gene expression. These data suggest that Foxi2 and Sox3 initiate and maintain significant co-localization on ectodermal CRMs during ZGA through the onset of gastrulation (S3B Fig).

Sox3 binds to 5,828, 4,402, and 14,648 peaks in stage 8, 9, and 10.5 embryos, respectively (Fig 3A). To gain further insight into Sox3 activity, we examined its overlap with Ep300, following the same approach used in our Foxi2 ChIP-seq analyses. Sox3 peaks display temporally unique (class I, III, VII), shared (class II, IV, VI), and persistently occupied (class V) binding patterns (Fig 3A and 3B). Class V, VI, and VII peaks all displayed higher Ep300 binding overlap and H3K27ac deposition compared to class I, II, III, and IV (Fig 3B and 3C), suggesting that class V-, VI-, and VII-associated genes are active. These results imply that, like Foxi2, Sox3 peaks that persist from stage 8 to 10.5 and 9 to 10.5, or are specif-ically bound at stage 10.5, are associated with Ep300 and represent active CRMs. We examined whether these CRMs are linked to transcriptional activity of the genes. Similar to Foxi2, Sox3 Class IV, V, VI, and VII associated genes are expressed at higher levels compared to class I, II, and III genes (Fig 3D).

Next, we further examined the genomic co-localization of Foxi2 and Sox3 in the context of ectoderm-specific Ep300 binding. Among the 4,866 Foxi2 and Sox3 early gastrula co-bound regions (Fig 2I), 2,901 also overlap with ectodermal Ep300 peaks (Fig 3E). These overlapping regions account for 95% of the 3,050 Foxi2/Ep300 peaks and 72% of the 4,015 Sox3/Ep300 peaks (Fig 3E). Additionally, these CRM regions show significantly higher Ep300 binding and H3K27ac mark deposition in the ectoderm compared to endoderm, suggesting their functional importance in ectodermal gene regulation (Figs 3F and S3B). Linking 2,901 Foxi2/Sox3 CRMs co-bound with Ep300 to their nearest genes, we find highly significant GO terms related to cell fate specification of neural (e.g., *fezf2, neurog3, zic3),* and epidermal lineages (e.g., *dlx3, grhl2, krt7*) (S3C Fig). This suggests that Foxi2 and Sox3 orchestrate a *cis*-regulatory code critical for driving both neural and non-neural ectoderm specification.

In early gastrula stage embryos, ectodermal genes such as *dlx5, foxi1, hgf, krt7, krt70, tfap2a,* and *tp63,* are bound by both Foxi2 and Sox3, and overlap with Ep300 peaks (Figs 3F and S3B). Interestingly, these CRMs are also premarked by Foxi2 and Sox3 by the 64-cell stage (Fig 2G). Given the established role of the Foxi family TFs in sensory placode forma-tion [28,29], we examined the key sensory placode genes, such as *eya1, six1,* and the anterior neural marker *fezf2,* which we found to be marked during cleavage stage (Fig 2G). Indeed, these genes are also co-bound by Foxi2 and Sox3 in the early gastrula stage embryos (S3B Fig), sharing the same CRMs pre-bound by Foxi2 and Sox3 during cleavage stages

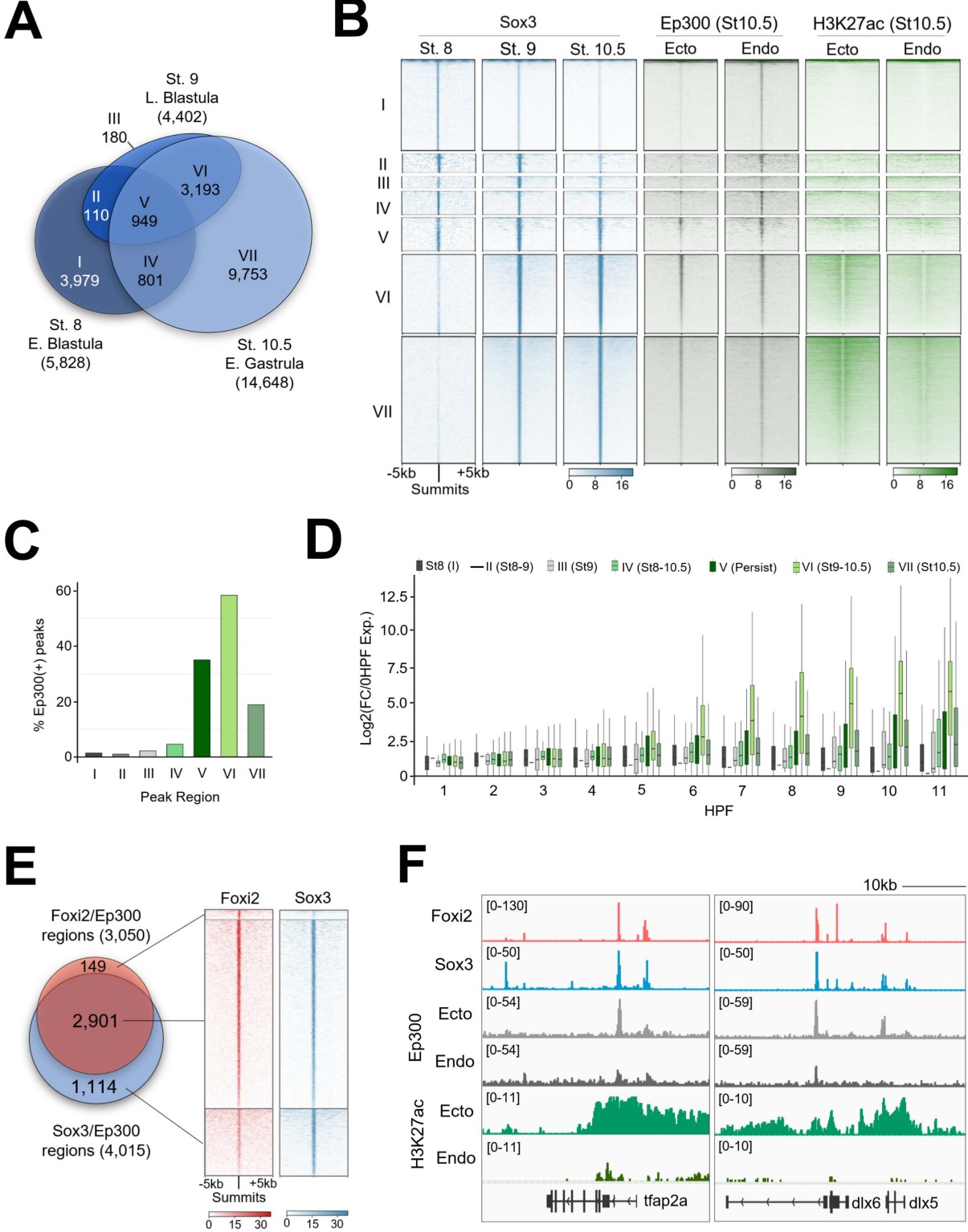

**Fig 3. Foxi2 and Sox3 preferentially colocalize at ectoderm CRMs in the presence of Ep300. (A)** Venn diagram illustrating the temporal dynamics of Sox3 ChIP-seq peaks, classified into categories I–VII. **(B)** Clustering analysis comparing Sox3 peak classifications with Ep300 and H3K27ac signals from early gastrula ectoderm and endoderm dissections. **(C)** Proportion of Sox3 class I–VII peaks overlapping with Ep300 peaks. **(D)** Temporal

gene expression profiles of genes located within 20 kb of Sox3/Ep300 co-bound peaks in Class I–VII. **(E)** Comparison of Foxi2 and Sox3 co-bound and independently bound regions overlapping with Ep300. **(F)** Genome browser tracks highlighting Foxi2, Sox3, and Ep300 binding along with H3K27ac enrichment at selected ectodermally expressed genes. The data underlying this figure can be found in S3 Data and GSE288637.

(Fig 2G). These findings underscore the master regulatory role of maternal TFs in pre-marking critical ectodermal CRMs for future gene expression.

### Independent and cooperative roles of Foxi2 and Sox3 in regulating ectodermal gene expression

To investigate the molecular function of Foxi2 and Sox3, we generated Foxi2 and Sox3 knockdown phenotypes using translation-blocking morpholino antisense oligonucleotides (MOs) (S4A and S4B Fig). Embryos injected with Foxi2 or Sox3 MOs showed substantial depletion of Foxi2 and Sox3 proteins (Fig 4A). While only a minority of Sox3 morphants exhibited decreases in body length and inhibited blastopore closure, Foxi2 knockdown embryos showed robust defects in both dorsoventral patterning and body length, as well as reduced head size (Fig 4B). RNA-seq analysis of both Foxi2 and Sox3 morphants compared to wild-type embryos identified genes activated and repressed by Foxi2 and Sox3 (Fig 4C and 4D). We intersected these data with Foxi2 and Sox3 ChIP-seq datasets to identify direct target genes, defined as those showing at least a 2-fold change in expression in the morphants and having Foxi2 and/or Sox3 binding sites within 20 kb of the gene (Fig 4C and 4D). At the late blastula stage, Foxi2 and Sox3 directly activate 78 and 114 genes, and repress 68 and 18 genes, respectively (Fig 4C). A similar analysis in gastrula-stage embryos revealed that Foxi2 and Sox3 directly activate 72 and 164 genes and repress 54 and 136 genes, respectively (Fig 4D). Intersecting the lists of genes activated by Foxi2 and Sox3 identifies 34 genes co-bound and co-regulated by both factors. These genes include well-studied ectodermal TFs such as *dlx3, dlx5, foxi1, hes5.1, tfap2a,* and *tp63* [30,31]. Reverse transcription (RT)-qPCR analysis of *foxi2* and *sox3* morphants confirms that these ectodermal TFs are jointly regulated by both factors (Fig 4E). Separately, 115 genes are regulated solely by Foxi2, while 243 genes are regulated solely by Sox3. Specific targets of Foxi2 include*, amotl1, dlc, gata3, kit, klf5,* and *krt70*, while direct targets of Sox3 include *cdx4, foxb1, foxd4l1.2, hes5.10, neurog3, olig4, sox21, zic3,* and *zic4.* These data suggest that Sox3 and Foxi2 regulate their specific target genes both jointly and independently, serving as master TFs that govern the early ectodermal program.

### Foxi2 and Sox3 directly co-regulate spatially distinct inner and outer layer cell states

To gain a deeper understanding of gastrula ectoderm differentiation than can be provided by bulk RNA-seq analyses, we assessed nascent transcripts at the single-cell level by performing single-nucleus (sn) RNA-seq on stage 10.5 wild-type embryos. We then characterized the cell-type-specific expression of Foxi2 and Sox3 direct target genes within this dataset. Pre-processing using a 1,500 genes/cell cut-off yielded 13,711 high-quality early gastrula nuclei (Fig 5A). De novo clustering and marker gene expression analysis identified a total of 13 defined cell states, consisting of 5 ectodermal, 4 mesodermal, and 4 endodermal cell types (Figs 5A, S5A and S5B). The five ectodermal cell clusters map at the bottom of two major lobes in the Uniform Manifold Approximation Projection (UMAP) (Fig 5A) and express markers that show these represent inner neural plate, inner neural plate border, inner non-neural ectoderm, outer neural ectoderm, and outer non-neural ectoderm. Notably, inner ectoderm cell clusters are positioned in the left lobe, while outer ectoderm cell clusters segregate in the right lobe (Figs 5A, S5C, and S5E). Two small lobes in the upper right (gray) contain unannotated cells (UC) that may represent a novel cell state characterized by *dscaml1* and *tp73* expression (S5D Fig) among others, but the identity of these cells is currently unclear.

After identifying 5 ectodermal cell clusters in stage 10.5 embryos, we mapped the expression patterns of Foxi2 and Sox3 direct target genes in the gastrula ectoderm to determine how Foxi2 and Sox3 contribute to the regional identity of this germ layer. There are 18 inner-layer and 14 outer-layer bound ectoderm target genes that are exclusively regulated

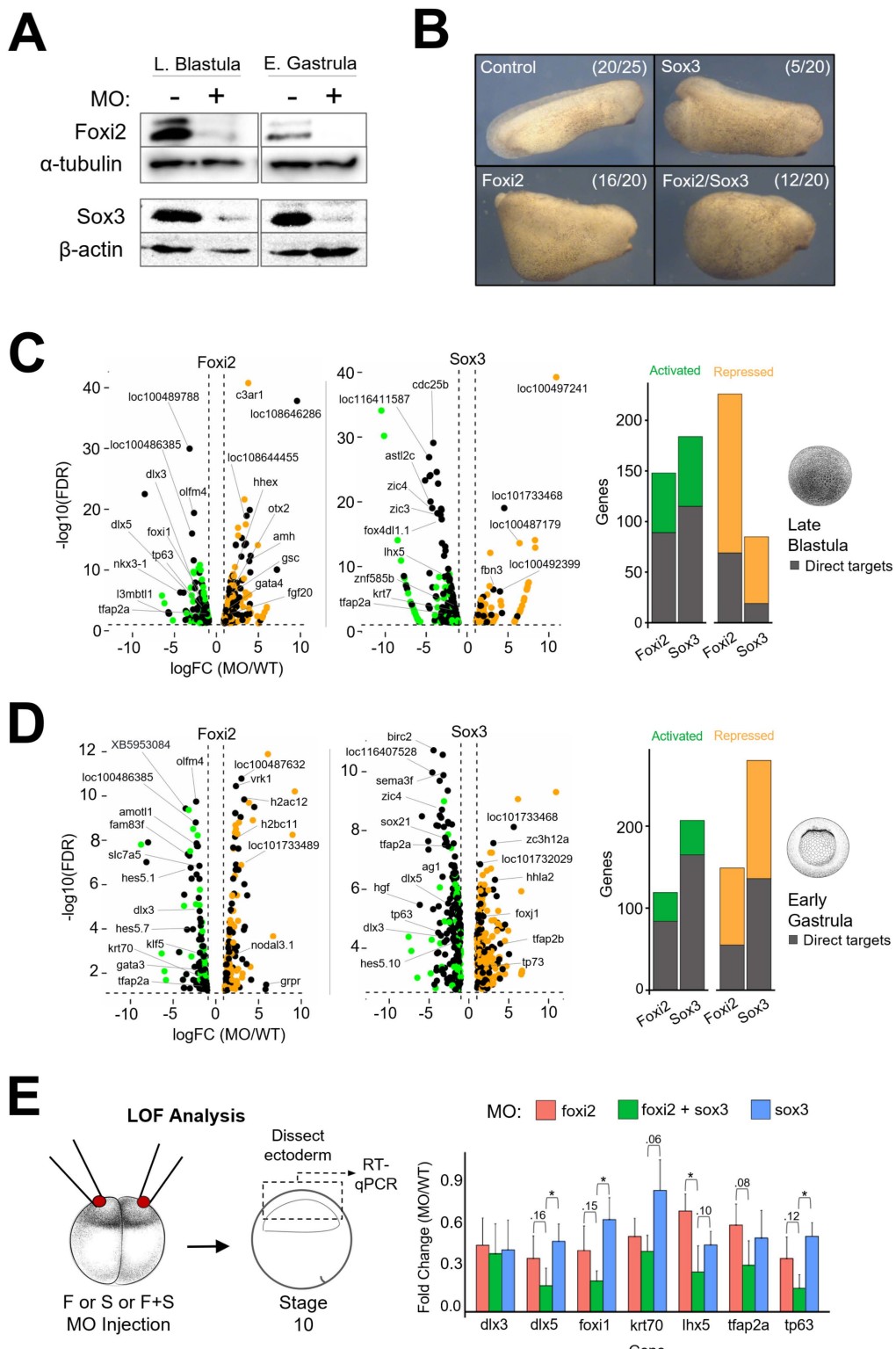

**Fig 4. Independent and cooperative roles of Foxi2 and Sox3 in regulating ectodermal gene expression. (A)** Western blot of Foxi2 and Sox3 morphant embryos at late blastula and early gastrula stages. **(B)** Phenotypic analysis of Foxi2, Sox3, and Foxi2/Sox3 morphants at the early tailbud stage. **(C, D)** RNA-seq analysis of late blastula embryos (top) and early gastrula embryos (bottom) showing activated and repressed genes within 20 kb

of Foxi2-/Sox3-bound regions. Histograms show the total number of direct target genes affected in morphants. **(E)** Schematic diagram of loss-of-function experiment (left). RT-qPCR analysis of key ectodermally expressed genes (right) in early gastrula ectoderm of Foxi2, Sox3, and Foxi2/Sox3 morphants. *eef1a1* expression was used for normalization. Error bars represent standard deviation from 3 biological replicates, asterisks indicate statistically significant differences ($p < 0.05$) determined by Student $t$ test. The data underlying this figure can be found in S4 Data and GSE288636. *Xenopus* illustrations © Natalya Zahn [27].

by Foxi2, which include *amotl1, klf5, krt70, lhx3, nkx3-1, and olfm4*. Additionally, 53 inner and 49 outer layer direct ectoderm target genes are exclusively regulated by Sox3, including *foxd4l1.2, neurog3, olig3, olig4, sal3, sox21, szl, zic3, and zic4,* many of which are involved in neurogenesis. Pan-ectodermal targets (e.g., *dlx3, hes5.1, klf5, lhx5,* and *tfap2a*) are expressed across all five cell states. Some of these genes are specifically regulated by Foxi2 or Sox3 alone, while others require the combinatorial input of both Foxi2 and Sox3 (Fig 5C). The observations suggest that Foxi2 and Sox3 regulate genes in all five distinct ectoderm cell states. Interestingly, we identified target genes of Foxi2 and Sox3 that are specifically expressed in the inner (both non-neural and neural) ectodermal layer, such as *foxi1, prdm14, szl, tp63,* and *zic4*. Similarly, genes such as *dlx5, hes4,* and *krt70* are expressed in the outer (both non-neural and neural) ectodermal cell states. Since Foxi2 and/or Sox3 regulate genes across both distinct ectodermal layers, we suggest that additional factors, like Prkci and Mark3, might play a role in differential segregation of these cell layers ([32,33], Tabler and colleagues [34]).

### Foxi2 and Sox3 are master regulators shaping the ectodermal epigenetic landscape

Given the potential synergy of Foxi2 and Sox3 within the ectoderm and their distinct roles in ectoderm target gene activation, could Foxi2 and Sox3 together serve as master regulators of ectodermal differentiation? To test this hypothesis, we injected Foxi2 and/or Sox3 mRNA into the vegetal pole and assessed the endoderm cell state in dissected vegetal tissue explants at gastrula stage 10.5 (Fig 6A). While overexpression of Foxi2 or Sox3 alone induced modest ectoderm gene expression, coinjection of both factors activated the ectodermal genes, *dlx5, foxi1, krt70, lhx5,* and *tfap2a*. Moreover, expression of key endodermal genes such as *gata4, gsc, hhex,* and *otx2* was significantly reduced in endodermal explants co-expressing Foxi2 and Sox3 mRNA. These findings support Foxi2 and Sox3 working together as master regulators of ectodermal identity.

In addition to the transcriptional role of Foxi2 and Sox3 in specifying the ectodermal cell state, we determined whether Foxi2 and Sox3 also shape the epigenetic landscape of ectodermal cells. Temporal analysis of H3K27ac and H3K4me1 signal deposition around 2,901 CRMs co-bound by Foxi2, Sox3, and Ep300 (FSE) (Fig 3E), reveals an increase from the early blastula to the early gastrula stage compared to independent Ep300 regions (Fig 6B). Interestingly, the repressive mark, H3K27me3 becomes gradually enriched around Ep300 peaks lacking Foxi2 and Sox3 binding (Fig 6B). These results suggest that Foxi2/Sox3 co-binding promotes H3K27ac and H3K4me1 enrichment at the expense of H3K27me3 deposition around ectodermal CRMs.

To overcome the lack of germ layer resolution in whole-embryo histone marking studies, we also performed experiments using dissected ectodermal and endodermal tissues from the early gastrula stage. While no significant differences in Ep300 binding were observed between ectoderm and endoderm at both Foxi2-Sox3 co-bound (FSE) regions and Ep300 only peaks (lacking Foxi2/Sox3) (Fig 6C), we found a significant difference in histone modifications. In the ectoderm, FSE regions showed significant accumulation of H3K27ac and H3K4me1, while H3K27me3 deposition is notably absent (Fig 6C). In contrast, Ep300-only peaks showed modest H3K27ac and H3K4me1 accumulation but displayed strong H3K27me3 enrichment. These findings suggest that Foxi2 and Sox3 specifically regulate the epigenetic landscape of ectodermal genes within the ectodermal germ layer, but not in the endoderm.

We sought to determine the role of Foxi2 and Sox3 in establishing the ectoderm-specific epigenetic landscape by measuring Ep300 genomic recruitment in the absence of both factors. To determine whether Foxi2 and Sox3 are required for tissue-specific Ep300 recruitment, we performed ChIP-seq on dissected ectoderm and endoderm from FS MO-injected

PLOS Biology

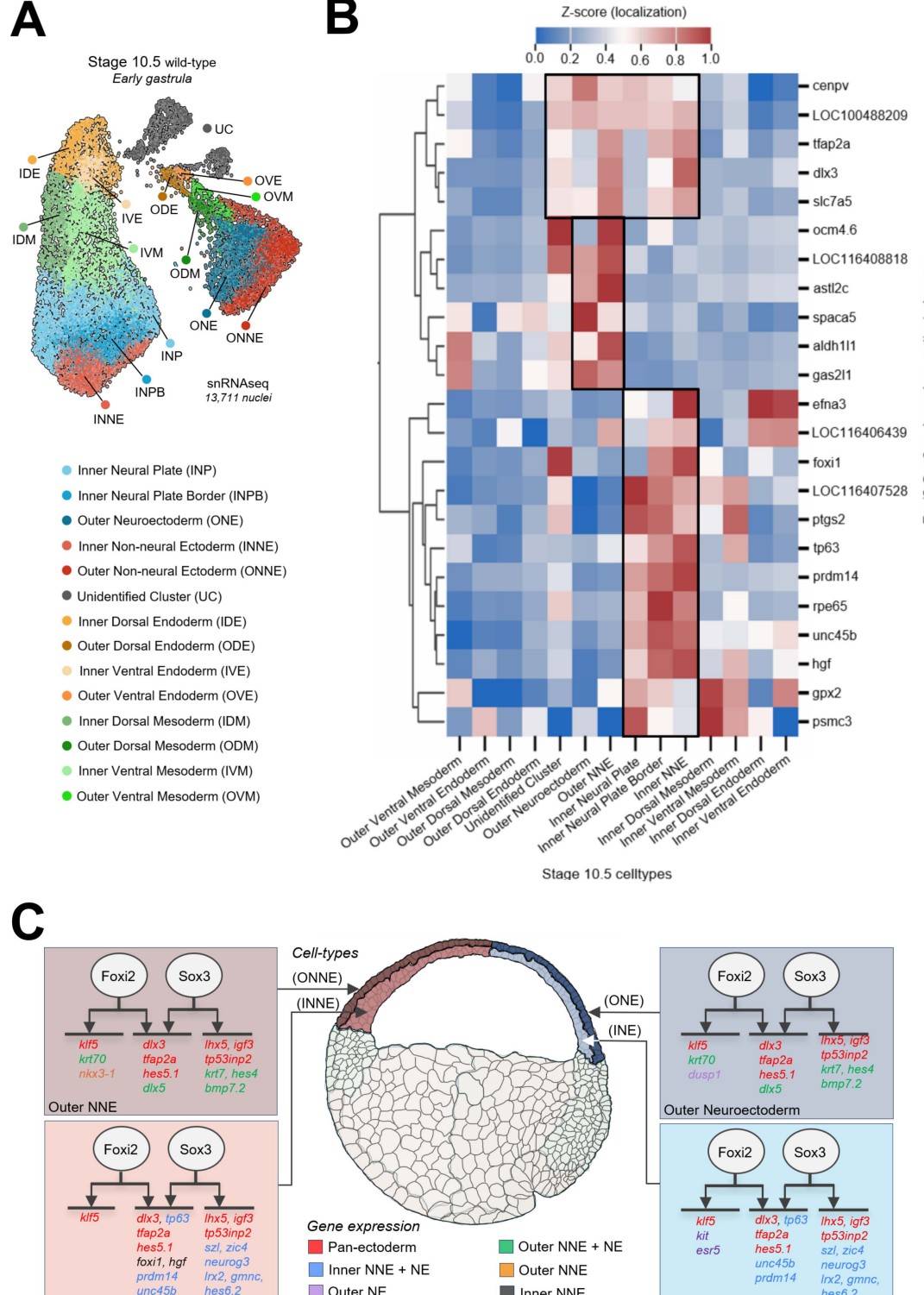

**Fig 5. Foxi2 and Sox3 directly co-regulate genes in all ectodermal cell states. (A)** Single nucleus RNA-seq UMAP (uniform manifold approximation projection) identifying 14 distinct cell types in the early gastrula. **(B)** Z-score expression analysis of Foxi2/Sox3 co-bound target genes shows distinct gene expression patterns marking outer and inner ectodermal cell types. The boxes represent genes expressed in both the inner and outer ectodermal

layers, as well as genes specific to either the outer or inner ectodermal layer. **(C)** Schematic diagram depicting the regulatory roles of Foxi2 and Sox3, either independently or jointly, in activating the expression of target genes across various ectodermal cell types. The data underlying this figure can be found in S5 Data and GSE288638.

and WT embryos (S6A Fig). Signal-based K-means clustering was performed to form FSE (I) and FSE (II) peak groups, which both display significant Ep300 deposition decreases in FS MO embryos compared to wild type (Fig 6D). In contrast, in FS MO embryos both non-FSE ectoderm Ep300, and vegetal Ep300-bound regions did not display significant decreases in Ep300 deposition (S6B Fig). Taken together, these data indicate that Foxi2 and Sox3 colocalization is required to recruit Ep300 to Foxi2 and Sox3-bound CRMs, comprising 49.7% of total ectodermal Ep300-bound regions in the early gastrula.

**Foxi2 and Sox3-marked regions establish ectodermal super enhancers to ensure robust gene expression**

The high levels of H3K27ac modifications around Foxi2 and Sox3 co-bound sites raise the question of whether some of these regions are located within SEs, which are large clusters of enhancers characterized by high levels of TFs, coactivators (such as Mediator), and chromatin modifiers. SEs are typically identified by analyzing genomic regions highly marked by H3K27ac [35]. H3K27ac ChIP-seq analysis of ectodermal and endodermal explants identified 14,088 ectoderm and 15,469 endoderm enhancer peaks ("regular" enhancers [REs]), with 6,962 being shared between the two germ layers (S6C Fig). Our investigation also identified SEs in both ectoderm and endoderm based on their increased size and H3K27ac deposition relative to the enhancer landscape (Figs 6E, S6C). The average size of both ectodermal and endodermal SEs is ~20kb per locus, which is significantly larger than RE loci, which are typically less than 2kb (S6D Fig). While 386 SE regions overlap between ectodermal and endodermal SEs, 761 are unique to the ectoderm, and 285 are unique to the endoderm (S6C Fig). These findings highlight that ectodermal SEs are more abundant than endodermal SEs at stage 10.5, and the timing of accumulation of H3K27ac suggests distinct germ layer SEs arise between the onset of ZGA and early gastrulation.

Highlighting the locations of genes associated with FSE-bound regions on a SE ROSE (ranked order of SEs) plot revealed that the Foxi2 and Sox3-regulated genes are highly enriched in ectodermal SEs relative to REs (Fig 6E–6G) or endodermal SEs (Fig 6G). The temporal expression levels of ectoderm SE genes associated with Foxi2-/Sox3-bound regions containing Ep300 (FSE) are also significantly higher than SE genes with no Ep300, and RE-associated genes (S6E Fig) during embryonic development. Interrogating our snRNA-seq data (Fig 5A), we found that ectodermal SE-associated genes are preferentially expressed in the ectodermal germ layer (Fig 6H), across ectodermal sub-lineages (S6F Fig). In contrast, endodermal SE-associated genes are preferentially expressed in the endodermal germ layer and its sub-lineages.

Finally, we investigated the gene expression variance (transcriptional "noise") of SE associated genes across individual cells. To quantify these variations in gene expression, we measured the coefficient of variation (COV), which reflects the extent of gene expression across individual cells within the dataset. Analysis of snRNA-seq data revealed that ectodermal SE-associated genes show higher gene expression levels and a lower COV, compared to RE genes and genes lacking H3K27ac deposition (S6G Fig). Additionally, we observed that SE-associated Foxi2/Sox3, Foxi2-only, and Sox3-only regulated target genes display the highest expression, and the lowest COV, compared to the entire landscape of both ectoderm SEs and REs, as well as genes without an active enhancer (Fig 6I). These results suggest that ectoderm SEs, enriched with Foxi2 and Sox3 binding, drive the most robust and stable expression of ectoderm-specific genes, while minimizing transcriptional noise. In sum, at the onset of gastrulation, Foxi2 and Sox3-marked regions preferentially acquire Ep300 and accumulate H3K27ac to establish SEs, which ensure robust gene expression across all ectoderm sub-lineages.

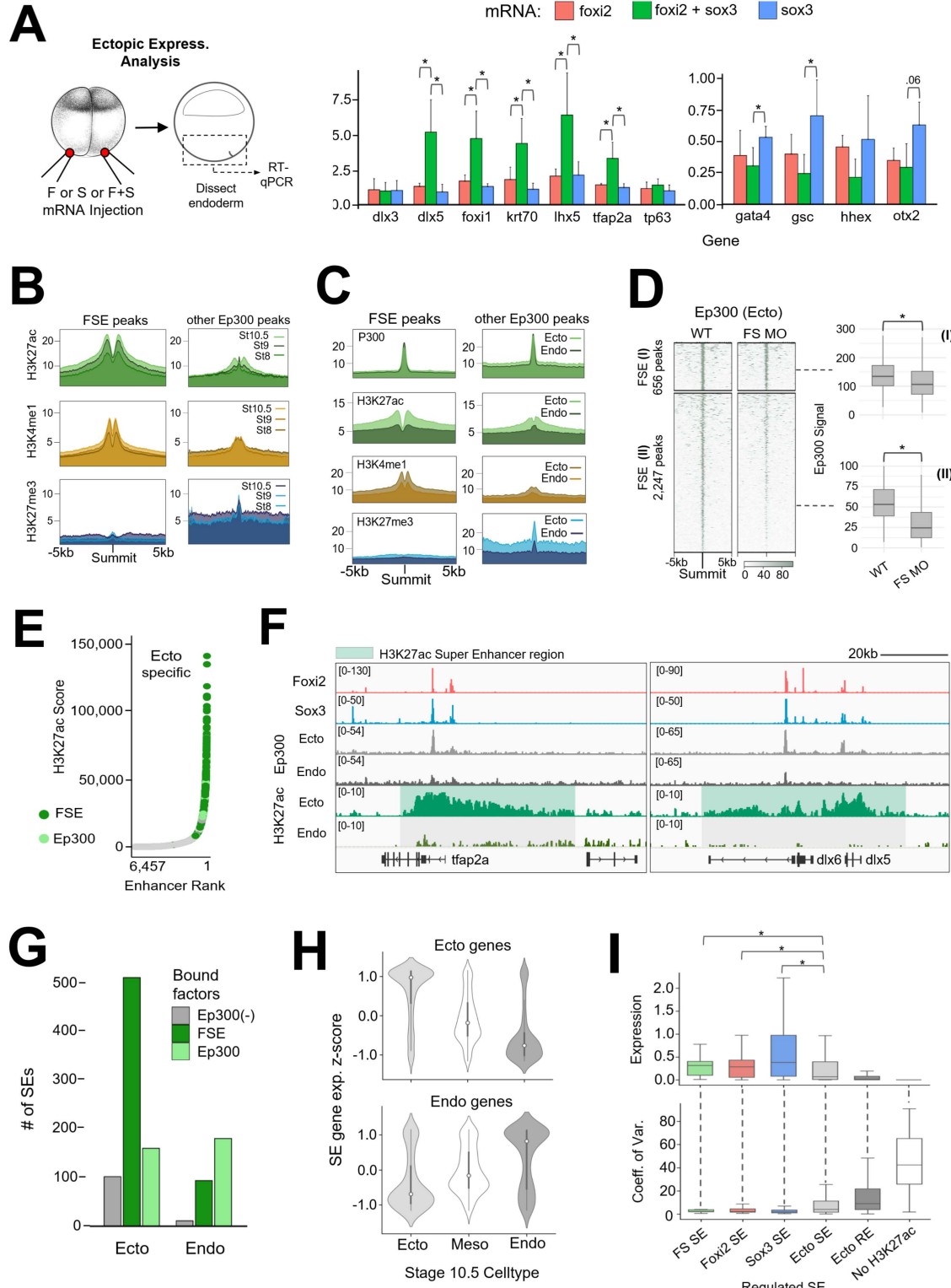

**Fig 6. Foxi2 and Sox3 are master regulators shaping the ectodermal super enhancer landscape. (A)** Schematic diagram of the experimental design (left). RT-qPCR analysis of key ectoderm and endoderm genes from early gastrula endoderm after ectopic expression of 1ng Foxi2, Sox3 or Foxi2/Sox3 mRNA. *eef1a1* expression was used for normalization. Error bars represent standard deviation from 3 biological replicates, asterisks

indicate statistically significant differences ($p < 0.05$) determined by Student $t$ test. **(B)** Accumulation of H3K27ac, H3K4me1, and H3K27me3 histone modifications around Foxi2/Sox3/Ep300 (FSE) co-occupied, and other solely bound Ep300 peaks. **(C)** Deposition of Ep300, H3K27ac, H3K4me1 and H3K27me3 at FSE and other Ep300 peaks present in the early gastrula ectoderm and endoderm. **(D)** Ep300 signal intensity in dissected ectoderm at enhancers co-bound by Foxi2, Sox3, and Ep300 (FSEs). Box-and-whisker plots show the distribution of Ep300 signals, highlighting differences between WT and FS morphants. **(E)** Total ectoderm SEs associated with FSE co-occupied peaks and Ep300 solely occupied peaks. **(F)** Genome browser views of ectodermal SEs (shaded area), which are present (highly acetylated) in ectoderm but absent in endoderm. **(G)** Histogram comparing the total number of SEs associated with peaks containing Foxi2/Sox3/Ep300 (FSE), Ep300 with insignificant Foxi2/Sox3 binding (Ep300), and without Ep300 (Ep300(−)). FSE peaks are highly associated with ectodermal SEs. **(H)** Z-score expression analysis of genes located within 20 kb of ectoderm (top) and endoderm (bottom) SEs. snRNA-seq cell-type expression analysis reveals that genes associated with ectodermal SEs are preferentially expressed in ectodermal cells. **(I)** Average expression levels (top) of Foxi2- and/or Sox3-regulated ectodermal SE-associated genes compared to all ectodermal SE-associated genes. Coefficient of variation (COV) analysis (bottom) using snRNA-seq shows that Foxi2 and/or Sox3-regulated ectodermal SE-associated genes exhibit lower expression variability than the full set of ectodermal SE-associated genes. In contrast, ectodermal RE-associated-gene display lower average gene expression and higher COV. Asterisks indicate statistically significant differences ($p < 0.05$) determined by Wilcoxon Rank-Sum test. The data underlying this figure can be found in S6 Data, GSE288637, and GSE288638. *Xenopus* illustrations © Natalya Zahn [27].

## Discussion

In this study, we identified maternal Foxi2 and Sox3 as master regulators of the ectodermal germ layer in *Xenopus*, based on the following evidence: (1) ectopic expression of these TFs in endodermal cells confers ectodermal cell fate specification; (2) loss-of-function analysis results in the loss of key ectodermal markers; (3) bound sites for these TFs are required for with ectodermal Ep300 recruitment to CRMs; and (4) the co-bound regions include ectoderm specific H3K27ac SEs that drive a robust ectodermal gene expression program. Since Foxi2 and Sox3 prebind to many CRMs that are critical for the later activation of ectodermal developmental programs (e.g., *eya1* and *six1* in sensory placode formation), we propose a model in which animally enriched Foxi2 and Sox3 establish early genomic occupancy at these CRMs during the cleavage stages. These interactions with the genome prefigure the accumulation of activating epigenetic modifications around these regulatory elements by early gastrulation, driving the formation of SEs and spatially distinct gene expression programs that define ectodermal identity.

### The roles of maternal Foxi2 and Sox3 in early development

In vertebrates, the Foxi subfamily comprises Foxi1, Foxi2, and Foxi3, which are pivotal in epithelial differentiation and organogenesis [36]. There are limited studies performed on the function of *foxi2* in early development. However, the available data consistently show that *foxi2* has roles at later stages, such as establishment of sensory placodes and anterior neural ectoderm. In the chick embryo, Foxi2 is crucial for craniofacial development, regulating the formation of the pharyngeal arches [37]. In mouse, Foxi2 is expressed in the developing forebrain, neural retina, dental and olfactory epithelium [38–40]. And in *Xenopus*, Foxi2 morphants similarly appear to have anterior structure defects (Fig 4B).

Despite these conserved roles of Foxi2 in post-gastrulation vertebrate embryos, evidence for maternally expressed Foxi2 being essential for early steps in ectodermal germ layer cell lineage development is currently limited to *Xenopus*. Maternally expressed Foxi2 directly activates *foxi1* (Figs 4C, 4E, and S4A; [25]), which is best characterized for its key roles in mucociliary development, ionocyte specification, and sensory placode formation in *Xenopus* [41]. This is consistent with the established roles of Foxi1 in other vertebrates. Interestingly, *Xenopus* Foxi2 pre-binds to the CRMs of *foxi1* as well as sensory placode gene *eya1* and anterior neural gene *six1* during cleavage stages (Fig 2G) and in early gastrulation (S3B Fig), perhaps priming these CRMs for subsequent activation during later stages of development.

Based on the evidence, we speculate that all three *Xenopus foxi* genes are involved in ectodermal specification, but their roles may have diverged during evolution. The maternal function of *foxi2* in early ectoderm specification may not be unique to early *Xenopus* embryos and might exist in other species, but hasn't yet been studied. In axolotl embryos, RNA-seq analysis shows that *foxi2* mRNA is detectable before zygotic transcription, suggesting it is maternally expressed in this urodele amphibian [42]. A zebrafish *foxi* gene, referred to as *foxi1*, and identified by our synteny analysis as

orthologous to *foxi2* in both *Xenopus* and mammals, is first expressed shortly after the onset of ZGA as indicated by RNA-seq profiling ([43]; https://www.ebi.ac.uk/gxa/home). This implies it might be more broadly expressed and have an early role in development in teleost fish. Finally, a single lamprey *foxi* gene is also expressed broadly in the animal pole at blastula stage [44]. These expression patterns suggest activity in the early ectoderm, implying conserved functions that pre-date the divergence of fishes [44]. In *Xenopus*, the *foxi1* and *foxi3* (formerly *foxi4*) genes are not maternally expressed but are transcribed after onset of ZGA. *foxi1* and *foxi3.2* (but not *foxi3.1*) are expressed broadly in the blastula ectoderm and later become confined to epidermal lineages, while being excluded from the neural plate [29,45,46]. Overexpression and loss-of-function experiments of various kinds suggest that these genes might have either redundant functions to *foxi2,* or more likely act as downstream effectors induced by *foxi2*. This is particularly the case with *foxi1.* Our data reveals that the early role(s) of *foxi2* in ectodermal germ layer specification includes activation of *foxi1*, however, *foxi3.2* does not appear to be regulated by Foxi2. Similar effects on ectodermal specification in other amniotes might involve a different timing, with target genes being induced by zygotically expressed *foxi2,* but further experiments in these species are needed to confirm this model.

In *Xenopus*, *sox3* expression is both spatially and temporally dynamic during development. *sox3* mRNA is maternally supplied with expression throughout the ectoderm prior to gastrulation and becomes restricted to the presumptive neural plate by mid-gastrula [26,47]. Sox3 represses endoderm formation to facilitate proper germ layer formation [26,48]. In later stages, *sox3* is expressed in neural progenitors along the neural tube, where it is implicated in promoting neural identity and preventing premature differentiation. Recently, Sox3 has been shown to have a role in ectoderm pluripotency, and also function as a pioneer TF affecting chromatin accessibility [49–51]. Our findings suggest that Sox3 has an additional role in orchestrating the ectodermal program during germ layer formation. This is supported by evidence showing that 1) several critical ectodermal genes are direct targets of Sox3, 2) Sox3 knockdown results in down-regulation of ectodermal gene expression, and 3) ectopic expression of Sox3 and Foxi2 upregulates ectodermal gene expression in the endoderm while repressing mesendodermal markers. It is plausible that Foxi2 and Sox3 primes pre-bound CRMs during early development and subsequently hands off its regulatory roles to other factors, such as *foxi1* or *soxE* family TFs [51]. This handoff mechanism could maintain CRM accessibility by preventing silencing through repressive histone modifications, thereby ensuring readiness for transcriptional activation in later developmental stages.

## Advantages of germ layer-specific SE formation in rapidly dividing embryos

We identified germ layer-specific SEs based on H3K27ac accumulation across the *Xenopus* embryonic genome. Notably, ectodermal SEs are highly correlated with the binding of Foxi2 and Sox3 TFs (Fig 6E—6G). Our findings also show that SE-associated Foxi2/Sox3 target genes exhibit higher expression levels compared to genes associated with REs, while displaying reduced expression noise (Fig 6I). Based on these germ layer-specific SE activities, we propose that SE-linked gene expression contributes to developmental stability during cell fate specification, thereby enhancing the robustness of embryonic tissue formation.

Rapidly dividing embryos, such as those of *Xenopus* and zebrafish, undergo swift transitions from gastrulation to neurulation within a short developmental timeframe. These aquatic embryos also face varying external influences, such as temperature, that impact their development. To ensure efficient and error-free progression, these embryos must not only achieve a rapid surge in gene expression during early stages but also tightly regulate expression to avoid critical errors. SEs offer a solution by concentrating TFs and cofactors at critical genomic regions, enabling robust and precise gene expression. We propose that in these rapidly dividing embryos, a key strategy might involve pre-loading enhancers with TFs before ZGA. While direct evidence from ChIP-seq at cleavage stages is lacking, limited ChIP-qPCR data suggest that pre-bound TFs can prime enhancers for rapid and robust activation at ZGA. At the onset of ZGA, recruitment of Ep300 and Kmt2c/d (the key histone-modifying enzymes responsible for H3K4 monomethylation) to CRMs, and RNA polymerase II to promoters may enable efficient transcription, ensuring developmental success in these time-constrained systems.

## Factors involved in segregating ectodermal cell clusters

The superficial (outer) and inner layer cells in *Xenopus* embryos have distinct transcriptomes in our snRNA-seq data (Figs 5 and S5). These differences arise due to distinct cues that are present by the 64- to 128-cell stages [32]. The localized expression of *tp63* in the inner (basal) layer and *hes5.10* and *krt70* in the outer layer has been detected through WMISH [52,53]. Additionally, our snRNA-seq analysis revealed enrichment of new markers, such as *col18a1* and *zeb2* in the inner layer, and *atp1b2, epas1,* and *nectin2* in the outer layer, among others, expanding our understanding of ectodermal lineage-specific gene expression patterns.

By examining the expression patterns of Foxi2 and Sox3 direct target genes in our UMAP of the early gastrula embryo (Fig 5), we discovered that these TFs coordinate the expression of pan-ectodermal genes. For example, genes such as *dlx3 and tfap2a* are co-regulated by both factors, while *klf5* expression is mainly regulated by Foxi2, and *igf3, lhx5,* and *tp53inp2* are regulated by Sox3. Since all these genes are expressed across the five ectodermal cell clusters (Fig 5B and 5C), we propose that Foxi2 and Sox3 exhibit both overlapping functions and distinct, non-overlapping roles in ectoderm development. Among Foxi2 and Sox3 targets, genes like *dlx5*, *hes5.10, krt7*, and *krt70* are exclusively expressed in the outer ectoderm, while *neurog3, prdm14, szl, unc45b,* and *zic4* and are specific to the inner ectoderm. This differential expression suggests that additional factors uniquely active or expressed in each layer may play a role. This highlights that other regulatory mechanisms are superimposed on Foxi2-/Sox3-regulated targets, which underlie this layer-specific regionalization of ectodermal cells occurring early in gastrulation.

Two notable candidates involved in this process are the kinases Prkci (also known as aPKC) and Mark3. Prkci is localized in apical cell membranes by the 4-cell stage [32] and contributes to the segregation of outer (superficial) ectodermal cells at later stages [33]. On the other hand, Mark3 is localized basally and influences the inner layer of cells [33]. This interplay may be essential for initiating outer and inner ectodermal differences that intersect with Foxi2/Sox3 regulated gene expression in the early *Xenopus* embryo.

Additional factors potentially involved in outer and inner ectoderm development include the Grainyhead-like (Grhl) TFs. Our motif analysis in regions bound by Ep300 revealed a significant enrichment of the Grhl motif in early gastrula ectoderm, but not in endoderm (Fig 1D). The *Grhl* gene family comprises three highly conserved members in vertebrates (*Grhl1-3*) [54–56]. Among these, *grhl1* is the only maternally expressed family member in *Xenopus*, which is also animally enriched [20] and continues to be expressed zygotically in both the outer and inner layer of the ectoderm based on our snRNA-seq data. Disruption of *grhl1* activity in *Xenopus* results in severe defects in epidermal differentiation, directly affecting keratin gene expression through binding of Grhl1 to the promoter region of *krt12.4* [57]. Similarly, *Grhl1* knockout models in both mouse and zebrafish exhibit disrupted epidermal cell differentiation (Wilanowski and colleagues, [58]; [59]). Both *grhl2* and *3* are zygotically expressed in *Xenopus,* starting during gastrulation and continuing into later stages [21,53]. *grhl2* is expressed in the inner layer while *grhl3* is expressed in the outer layer of gastrula ectoderm, and later in the superficial layer of epidermis [53]. Our snRNA-seq data confirms these *grhl* expression domains. In mice, loss of *Grhl2* causes non-neural ectoderm disruption, leading to disorganized cell junctions, aberrant mesenchymal protein vimentin, and decreased epithelial integrity [60]. *Grhl3*-deficient mouse embryos die shortly after birth due to impaired skin barrier function [61]. We observed that a Grhl motif is not enriched in Foxi2- and Sox3-bound regions (Fig 1D), suggesting that *grhl1* likely regulates a distinct set of ectodermal target genes compared to those of Foxi2 and Sox3. We propose that Grhl1 functions as a maternal ectodermal TF that functions in parallel with Foxi2 and Sox3 to initiate the global ectodermal specification programming, potentially through a separate pathway. Additionally, *grhl2* and *grhl3* may take over zygotic functions of *grhl1*, operating within the two ectodermal layers during later stages of development. Our future goals focus on dissecting the regulatory pathways of the ectoderm by combining loss-of-function analysis of maternal TFs with single-nucleus transcriptomics. This approach aims to determine the timing of lineage segregation and uncover the contributions of the maternal TFs to the specification of various ectodermal cell lineages and epigenetic changes.

## Ethics statement

Animal protocols were approved by the Institutional Animal Care and Use Committee (IACUC) of University of California, Irvine, under Protocol No AUP-24-067. No surgical procedures were performed. All handling and experimental procedures were conducted in accordance with institutional and national guidelines to ensure animal welfare, and efforts were made to minimize discomfort and stress.

## Methods

### Chromatin immunoprecipitation (ChIP) assays

Antibodies for Foxi2 [25] and Sox3 [Zhang and colleagues [62]] were validated previously, while the Ep300 antibody was from Santa Cruz sc-585, the H3K27me3 antibody from Upstate/Millipore 07-449, and the H3K27ac antibody from Abcam ab4279. ChIP-qPCR was performed as described in Chiu and colleagues [63], using whole embryos or dissected tissue fragments with specific primer sets. ChIP-seq libraries were generated using NEB E7645S DNA sequencing kit or the Bioo Scientific NEXTflex ChIP-seq kit, verified using an Agilent Bioanalyzer 2,100, and sequenced at UC Irvine's Genomics Research and Technology Hub using the Illumina Novaseq platform.

### Western blot

Embryos were homogenized in 1×RIPA buffer (50 mM Tris-HCl pH 7.6, 1% NP40, 0.25% Na-deoxycholate, 150 mM NaCl, 1 mM EDTA, 0.1% SDS, 0.5 mM DTT) supplemented with Roche's cOmplete protease inhibitor. The homogenate was centrifuged at 14,000 rpm, and the supernatant was collected and centrifuged again. Western blotting was performed on the final supernatant using anti-Foxi2, anti-Sox3, alpha-tubulin (Sigma, T6168), or beta-actin antibodies (Sigma, A5316).

### RNA assays

Total RNA was extracted from embryos using TRIzol reagent (ThermoFisher) [64]. Polyadenylated mRNA transcripts were isolated using NEBNext PolyA mRNA Magnetic Isolation Module (NEB E7490S). RNA libraries were prepared using NEBNext Ultra II Library Preparation Kit (NEB E7770S), validated using an Agilent Bioanalyzer 2100, and sequenced on the Illumina Novaseq platform. RT-qPCR analyses were performed using primers listed in the S12 Data supplementary file. Reverse transcription was performed using the MMLV reverse transcriptase (ThermoFisher Superscript II). qPCR was carried out on a Roche Lightcycler 480 II using Roche SYBR green I master mix.

### Morpholino knockdown, rescue, and vegetal RNA injections

*foxi2* translation blocking morpholino was created by GeneTools, 5′-TTATGAAGTCTGGTGGGACATTCAC-′3. Morpholino rescue was performed using mRNA prepared from a pCS2+ FLAG-*foxi2* construct. *sox3* translation blocking morpholino is 5′-GTCTGTGTCCAACATGCTATACATC-3′. *sox3* morpholino rescue was performed using a pCS2+ FLAG-*sox3* construct. The *X. tropicalis foxi2* coding sequence was acquired from the *X. tropicalis* Unigene library (TGas144f13) in the pCS107 vector, and the *X. tropicalis sox3* coding sequence was cloned into the pCS2+ vector. The *foxi2* plasmid was linearized with HpaI, and the *sox3* plasmid with XbaI, for generating capped mRNA using the SP6 mMessage Machine kit. Morpholino knockdowns were performed by injecting directly into opposing sides of the animal cap at the 1-cell stage or into each animal blastomere at the 2-cell stage. Animal caps were dissected one hour prior to harvesting for RNA preparation. Vegetal RNA injections were performed using indicated amounts of mRNA per embryo into opposing sides of the vegetal mass at the 1-cell stage, or in each vegetal blastomere at the 2-cell stage. Vegetal masses were dissected at blastula stage ~1 hour before harvesting with Trizol.

## Nuclei preparation and snRNA sequencing

Nuclei were isolated from stage 10.5 embryos using previously published discontinuous sucrose gradient protocols followed by a separate centrifugation through 80% glycerol [65–67]. Briefly, 200 embryos were homogenized in a 250 mM sucrose solution containing glycerol and snap frozen in liquid nitrogen [65]. Homogenates were later thawed on ice, brought to ~2.2 M sucrose and centrifuged through a 2.4 M sucrose layer at 130,000*g* for 2 hours at 4 °C using a Beckman SW55Ti rotor, followed by a 3400*g* centrifugation for 10 min at 4 °C through an 80% glycerol cushion. RNAse (NEB) and protease inhibitors (Roche 7× cOmplete, Mini, EDTA-free) at 0.2 U/ml and 5 mg/ml, respectively, were used throughout the previous steps. Nuclei were finally resuspended in "nuclear PBS" (0.7× PBS, 2mM MgCl2) containing 0.5% BSA and 0.2 U/ml RNase inhibitor. DAPI-stained nuclei were counted using a hemocytometer, and then were subject to fixation using the Parse Bioscience Evercode Fixation Kit (SB1003). Nuclei in DMSO were slow-frozen in a −80 °C overnight according to the kit manual. A bar-coded snRNA-seq library was prepared using the Parse WT Mini Kit (ECW01010) and sequenced at the University of California, Irvine Genomics Research and Technology Hub.

## Chromatin immunoprecipitation analysis

**ChIP-qPCR:** Percent input was calculated according to [Lin and colleagues [68]], where Input % = 100/2^([Cp[ChIP] − (Cp[Input] − Log2(Dilution Factor)). **ChIP-seq peak calling and IDR**: Reads were aligned to the *X. tropicalis* genome V10.0 using Bowtie 2 v2.4.1 [69] using default options. The.sam alignment was converted to.bam, duplicates were removed and files were converted to.bed format using samtools [Li and colleagues [70]]. Peaks were called against stage-specific (this paper and [18]) and tissue-dissected input DNA using MACS2 v2.2.7.1 [71] with the "-p.001" argument as the only non-default option. For narrowPeaks, Irreproducibility discovery rate was followed according to [72], where optimal peaks were selected using 0.05 *p*-value threshold. **Heatmap generation**: Bigwig files were created using "bamCoverage" where "—scale-factor" was used to evenly scale reads between ChIP-seq experiments using the same antibody, and RPKM normalized using DeepTools [73]. "computeMatrix" was used to generate a matrix by mapping.bigwig coverage files to.bed peak regions, then "plotHeatmap" was used to visualize the matrix. Average signal intensity profiles (Figs 2E, S1B, and 5A) were made from matrices using "plotProfile". **Motif analysis:** HOMER (V4.11) [74] was used to predict de novo motifs using the "findMotifsGenome.pl" command within 100 bp of the peak summit. To analyze motif occurrences within Foxi2 peaks, the following command was used: "annotatePeaks.pl $PEAKS $GENOME -size 2000 -hist 20 -ghist -m $MOTIF -mbed $MOTIF_BED> $OUTPUT". The resulting matrix was visualized in Python using numpy (v1.26.2 [75]) and matplotlib (v3.8.2 [76]). **Peak location annotation**: To categorize whether a peak fell within a particular genomic region "annotatePeaks.pl $PEAKS $GENOME_FASTA -gtf $GFF3> $OUTPUT" was used. **Genome browser visualization**: After alignment,.bam files were converted to.bed files used samtools command 'bamtobed'. HOMER's "makeTagDirectory" was then used to generate a tag directory as an input for "makeUCSCfile" which outputs a bedgraph. The Integrative Genome Viewer was then used to convert the bedgraph into a.tdf file using igvtools "toTDF". **Enhancer ranking**: The rank-ordered super enhancer (ROSE) from the Young lab [35] was used to categorically assign epigenetic peaks as REs or SEs based on an enhancers overall length and signal density. SEs were generated by stitching adjacent enhancer loci when loci passed a signal threshold. **Published ChIP-seq datasets:** Ep300 Stage 9, 10.5 [19], H3K4me1 Stage 8 [50], Stage 9, 10.5 [19], H3K27ac Stage 8, 9, 10.5 [16]. Animal and vegetal H3K4me1 at stage 10.25 [77].

### Quantification and statistical analysis

**RT-qPCR:** The ΔΔCp method [78] was utilized for calculating the fold-change in gene expression between treatment and control where the error among biological replicates was calculated using standard deviation and significance using a two-tailed *t* test. Bulk RNA-seq: Reads were aligned using RSEM V1.3.3 [79] with STAR V2.7.3 [80] to the *Xenopus tropicalis* V10.0 genome to generate normalized expected read counts. Differential gene expression analysis was performed in R

V4.1.1 using DEseq2 V1.34.0 [81]. **Localized gene expression analysis:** Early gastrula tissue dissection RNA-seq data from [22] was used to determine ectoderm and endoderm localized genes. When comparing vegetal mass and animal cap dissections, genes with a 2-fold enrichment in either germ layer were categorized as local, then the top 250 zygotic genes ranked by false discovery rate were extracted. **Zygotic gene expression timecourse:** Temporal RNAseq data from [21] was used to generate expression timecourses for zygotic gene sets. The 0HPF time point was used for detecting maternal transcript expression (at least 1TPM) to segregate zygotically expressed genes for analysis of downstream timepoints. 0HPF time point TPMs were compared with downstream time point TPMs to calculate zygotic gene expression log fold-changes.

### snRNA-seq analysis

**Alignment:** Parse Biosciences' in-house alignment software (dnanexus.com) command "Parse Batch Analysis v1.1.4" was used with the default commands against the UCB_Xtro_10.0 (GCA_000004195.4) reference. **Formatting:** The unfiltered Parse alignment output containing "all_genes.csv", "cell_metadata.csv" and "DGE.mtx" were first amalgamated into a.h5Seurat object using the R v4.3.1(http://www.r-project.org) software Seurat v4.3.0 [82], Matrix v1.6.5, SeuratDisk v0.0.0.9015 (https://github.com/mojaveazure/seurat-disk), Reticulate v1.37.0 [83] and the command "CreateSeuratObject". Finally, the.h5Seurat object was converted into an.h5ad file using the "Convert" command with the option "dest = "h5ad"". **Pre-processing:** Using scanpy v1.9.6 [84], pandas v2.1.4 [85], the nuclear data was log transformed and normalized and highly variable genes were annotated. Doublets were predicted using Scrublet [86], then manually inspected, confirmed, and removed based on illogical ectopic co-expression of germ-layer specific markers and outlier read counts. **Cell lineage annotation:** Leiden [87] was used to predict de novo cluster formation, visualized via UMAP [88]. Scanpy was used to calculate differential gene expression analysis of known marker genes between de novo clusters which were assigned to their germ-layer identity. Accordingly, clusters were merged based on co-expression of known marker genes. In this way, cluster boundaries are initially formed in an unbiased way through de novo clustering, based on differential gene expression. In-situ hybridization studies of marker genes were subsequently used for appropriate cluster annotation and merger. **Gene expression analysis:** Pandas was used for both SE genes', and Foxi2/Sox3-associated genes' *z*-score localization calculations by examining gene expression within the cells from each cell-type annotation, or merged annotations. Coefficient of Variation (COV) and average expression were calculated using numpy v1.26.2 [75].

## Supporting information

**S1 Fig. Differential epigenetic marking of ectodermal and endodermal cells. (A)** H3K27ac ChIP-seq analysis of ectodermal and endodermal explants reveals both lineage-specific and shared peaks, indicating distinct and overlapping enhancer activity. **(B)** H3K27me3 peak distribution in ectodermal and endodermal tissues. **(C)** Genome browser views showing H3K27ac and H3K27me3 peaks at ectodermally expressed genes (*grhl3, krt7, krt70*) and endodermally expressed genes (*vegt, foxa1, gsc*). **(D)** Distribution of Ep300 peaks in ectodermal and endodermal explants. The data underlying this figure can be found at GSE288637.
(TIFF)

**S2 Fig. Temporal Foxi2 binding and epigenetic marking deposition. (A)** Time course of mRNA expression for *ep300, foxi2*, and *sox3*. **(B)** Average signal density plots showing enrichment of H3K27ac and Ep300 peaks around Foxi2 binding sites in ectodermal and endodermal tissues, based on clusters I–VII defined in Fig 2A and 2B. **(C)** Venn diagram illustrating the overlap of peaks among Foxi2, Ep300, and H3K27ac in early gastrula-stage embryos. The data underlying this figure can be found in S7 Data and GSE288637.
(TIFF)

**S3 Fig. Co-occupancy of Foxi2 and Sox3 on the embryonic genome. (A)** Overlap of Sox3 and Foxi2 ChIP-seq peaks in stage 8, 9, and 10.5 embryos, indicating dynamic co-binding across developmental timepoints. **(B)** Genome browser views showing Foxi2 and Sox3 binding near ectodermally expressed genes across multiple stages, alongside with Ep300, H3K27ac, and ATAC-seq profiles from dissected ectodermal and endodermal explants. **(C)** Gene ontology (GO) enrichment analysis of Foxi2/Sox3 co-bound peaks that are also marked by Ep300. The data underlying this figure can be found in S8 Data and GSE288637.
(TIFF)

**S4 Fig. Foxi2 and Sox3 morpholino knockdown specificity. (A)** Alignment of the *foxi2* morpholino (MO) sequence with the wild-type target site and the rescue mRNA 5′ coding sequence used in the rescue experiment. A schematic illustrates the isolation of ectodermal tissues from wild-type, morpholino-injected, and rescued embryos. The bar graph shows fold changes in the expression of Foxi2 direct target genes in *foxi2* morphants. **(B)** Alignment of the *sox3* MO sequence with the wild-type *sox3* gene and the corresponding rescue 5′ mRNA coding sequence overlapping the MO target site. The bar graph shows fold changes in the expression of Sox3 direct target genes in *sox3* morphants. The data underlying this figure can be found in S9 Data. *Xenopus* illustrations © Natalya Zahn [27].
(TIFF)

**S5 Fig. De novo clustering of snRNA-seq from early gastrula embryos. (A)** Heatmap showing representative marker genes across 13 identified clusters and one newly identified cluster. **(B)** UMAP representation of all cell clusters identified in stage 10.5 embryos as reference. **(C–G)** UMAP plots showing co-expression of selected genes within specific clusters, highlighting cell-type-specific transcriptional profiles and validating the locations of clusters in panel B. The data underlying this figure can be found in S10 Data and GSE288638.
(TIFF)

**S6 Fig. Foxi2 and Sox3 co-bound regions define ectodermal SEs with high and stable gene expression. (A)** Genome browser views of Ep300 binding in ectodermal and endodermal tissues from *foxi2/sox3* (FS) morphants, and wild-type (WT) embryos. **(B)** Ep300 signal intensity in non-FSE (Ep300 unique) peaks in dissected ectoderm (top) and Ep300 signal intensity across all Ep300 endoderm peaks in dissected endoderm (bottom). Box-and-whisker plots show signal distributions are not statistically significantly (n.s.) different between WT and FS morphants. **(C)** Identification of ectodermal and endodermal SEs using the ROSE algorithm, based on H3K27ac ChIP-seq signal enrichment. Venn diagram shows SE and RE overlaps between ectoderm and endoderm. **(D)** SE-associated genes are highly expressed in their respective germ layers compared to RE-associated genes. **(E)** FSE-associated genes display significantly higher temporal expression than SE genes lacking Ep300 (P(−)SE) or regular enhancer (RE)-associated genes. **(F)** Ectodermal and endodermal SE-associated genes are preferentially expressed in the respective ectodermal and endodermal sublineages. **(G)** SE-associated genes exhibit higher expression levels and lower coefficient of variation (COV) compared to RE-associated genes and genes without H3K27ac enrichment. The data underlying this figure can be found at S11 Data, GSE288637, and GSE288638.
(TIFF)

**S1 Data. Dissected epigenetic peaks associated with top ectoderm and endoderm localized zygotic genes and their gene coordinates.**
(XLSX)

**S2 Data. Foxi2 Class I–VII peak regions and their overlap with ectoderm dissected Ep300.** Time-course matrices of Foxi2/Ep300-associated activated genes. Fox, Sox, Pou motifs occurring within Foxi2/Ep300 (Ecto) shared peak regions. Foxi2, Sox3, Ep300 64-cell stage ChIP-qPCR matrix. Foxi2 and Sox3 St. 10.5 shared and unique peaks.
(XLSX)

**S3 Data. Sox3 Class I–VII peak regions and their overlap with ectoderm dissected Ep300.** Time-course matrices of Sox3-/Ep300-associated activated genes. Foxi2, Sox3, and Ep300 (Ecto) St. 10.5 shared peak regions.
(XLSX)

**S4 Data. Foxi2 and Sox3 binding and localization annotated differential gene expression lists.** Foxi2 and Sox3 co-knockdown RT-qPCR matrix.
(XLSX)

**S5 Data. Foxi2 and Sox3 co-bound, co-activated ectoderm localized *z*-score matrix, and differential gene expression.**
(XLSX)

**S6 Data. Foxi2 and Sox3 ectopic expression RT-qPCR matrix.** Ep300 (Ecto) peaks shared with, and unique from, Foxi2/Sox3 bound regions. Foxi2, Sox3, and Ep300 (Ecto) class I, II regions with associated.bigwig comparison matrix. Dissected ecto- and endoderm ranked H3K27ac enhancer regions overlapping with Foxi2, Sox3, and Ep300 peaks. Ectoderm super enhancer (SE) associated gene's average expression and coefficient of variation matrices.
(XLSX)

**S7 Data. Foxi2, Sox3, and Ep300 time-course expression matrix and Foxi2 Class I–VII peak regions.**
(XLSX)

**S8 Data. Foxi2 St. 8, 9, 10.5 peak regions.** Sox3 St. 8, 9, 10.5 peak regions. Foxi2, Sox3 shared activating peak Gene Ontology matrix.
(XLSX)

**S9 Data. Foxi2 and Sox3 rescue matrices.**
(XLSX)

**S10 Data. St. 10.5 lineage marker expression z-score matrix derived from snRNA-seq.**
(XLSX)

**S11 Data. Ep300 (Ecto) regions unshared with Foxi2, Sox3 with associated.bigwig comparison matrix.** Ep300 (Endo) regions with associated.bigwig comparison matrix. Regular and super enhancer length comparison matrix. Super enhancer-associated gene expression time-course matrices. Ecto- and endoderm super-enhancer-associated gene localization *z*-score matrices. Ecto- and endoderm enhancer-associated gene's average expression and coefficient of variation matrices.
(XLSX)

**S12 Data. Primer sequences for RT-qPCR analyses.**
(XLSX)

**S1 Raw Images. Supplemental Data.** *Validation of Foxi2 Protein Knockdown*. Western blot was performed on Stage 9 and 10.5 embryos injected with Foxi2 morpholino at the 1–2 cell stage. Foxi2 (left) and alpha-Tubulin (right) expression are compared between wild-type and morpholino-treated embryos. The two raw exposures used for the Foxi2 and alpha-Tubulin bands in the main figure text are presented in the middle and bottom figures.
(PDF)

**S2 Raw Images. Supplemental Data.** *Validation of Sox3 Protein Knockdown.* Western blot was performed on Stage 9 and 10.5 embryos injected with Sox3 morpholino at the 1–2 cell stage. Sox3 (bottom) and beta-Actin (top) expression are

compared between wild-type and morpholino-treated embryos. The Sox3 (bottom) and beta-Actin (top) are the raw exposures used in the main figure text, however, the columns marked "x" were not used to make the publication figure. (PDF)

## Acknowledgments

This work was made possible, in part, through access to the University of California, Irvine Genomics Research and Technology Hub (GRT Hub) parts of which are supported by NIH grants to the Comprehensive Cancer Center (P30CA-062203) and the UCI Skin Biology Resource Based Center (P30AR075047). We thank Xenbase (http://www.xenbase.org/, RRID: SCR_003280) and the National *Xenopus* Resource (RRID:SCR013731), for genomic and community resources, and the University of California, Irvine High Performance Computing Cluster (https://hpc.oit.uci.edu/) for their valuable resources and helpful staff.

## Author contributions

**Conceptualization:** Clark L. Hendrickson, Ira L. Blitz, Ken W. Y. Cho.

**Data curation:** Clark L. Hendrickson, Ira L. Blitz, Kitt D. Paraiso, Jin S. Cho, Michael W. Klymkowsky, Matthew J. Kofron.

**Formal analysis:** Clark L. Hendrickson, Amina Hussein.

**Funding acquisition:** Ken W. Y. Cho.

**Investigation:** Ira L. Blitz, Ken W. Y. Cho.

**Project administration:** Ken W. Y. Cho.

**Supervision:** Ira L. Blitz, Ken W. Y. Cho.

**Validation:** Clark L. Hendrickson.

**Writing – original draft:** Clark L. Hendrickson, Ira L. Blitz, Ken W. Y. Cho.

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
