## [Editor Report · Decision Letter 0]

8 Mar 2025

Dear Dr Cho,

Thank you for submitting your manuscript entitled "Foxi2 and Sox3 are master regulators controlling ectoderm germ layer specification" for consideration as a Research Article by PLOS Biology.

Your manuscript has now been evaluated by the PLOS Biology editorial staff as well as by an academic editor with relevant expertise and I am writing to let you know that we would like to send your submission out for external peer review.

Once your full submission is complete, your paper will undergo a series of checks in preparation for peer review. After your manuscript has passed the checks it will be sent out for review. To provide the metadata for your submission, please Login to Editorial Manager (https://www.editorialmanager.com/pbiology) within two working days, i.e. by Mar 11 2025 11:59PM.

Kind regards,

Ines

--

Ines Alvarez-Garcia, PhD

Senior Editor

PLOS Biology

---

## [Decision Letter · Decision Letter 1]

7 May 2025

Dear Dr Cho,

Thank you for your patience while your manuscript entitled "Foxi2 and Sox3 are master regulators controlling ectoderm germ layer specification" was peer-reviewed at PLOS Biology. The manuscript has now been evaluated by the PLOS Biology editors, an Academic Editor with relevant expertise, and by two independent reviewers.

The reviews are attached below. As you will see, the reviewers find the conclusions novel and interesting, however they also raise several points that would need to be addressed before we consider the manuscript for publication. Reviewer 1 is not convinced by the conclusions for genes that seem to be expressed many hours before the ChIP-seq data provided in later blastula, and asks for several clarifications on the results. In addition, this reviewer suggests rearranging some of the supplementary figures into the main ones to improve the flow. Reviewer 2 proposes an experiment performing Ep300 ChIP after Foxi2 and Sox3 knockdown in ectoderm vs endoderm explants to confirm that Foxi2 and Sox3 are required for Ep300 to bind specific regions in the ectoderm and modify histones to activate gene expression.

In light of the reviews, we would like to invite you to revise the work to thoroughly address the reviewers' reports.

Given the extent of revision needed, we cannot make a decision about publication until we have seen the revised manuscript and your response to the reviewers' comments. Your revised manuscript is likely to be sent for further evaluation by all or a subset of the reviewers.

**IMPORTANT - SUBMITTING YOUR REVISION**

3. Resubmission Checklist

a) *PLOS Data Policy*

b) *Published Peer Review*

Sincerely,

Ines

--

Ines Alvarez-Garcia, PhD

Senior Editor

PLOS Biology

Reviewers' comments

Rev. 1:

In this interesting and experiment-rich manuscript the authors investigate determinants of cell fate specification in early embryo development. They identify Foxi2 and Sox3 as master regulators of ectodermal gene expression and show how epigenome remodeling is governed by binding of these transcription factors binding, p300 recruitment and H3K27 acetylation. The authors first characterize chromatin remodeling around ectoderm and endoderm genes following genome activation and at the onset of germ layer specification using ChIP-seq of segmented tissue from the Xenopus early gastrula. They reveal germ layer specific increases in H3K27Ac around ectoderm and endoderm genes and reciprocal down-regulation of ectoderm genes via H3K27me in endoderm cells. This highlights both positive and negative contributions to cell fate determination and how ectoderm expression is kept off in endoderm. Similarly they find p300 enriched proximal to ectoderm genes - but not endoderm genes - in ectoderm cells. Motif analyses of p300 local gene peaks revealed Sox and Fox binding sites in ectoderm genes, which provide the rationale for experiments throughout the remainder of the study. Next, they quantify binding of Foxi2 and p300 as well as deposition of H3K27Ac throughout the genome from mid-blastula to early gastrula. They create unbiased categories of Foxi2 binding and demonstrate a strong relationship between Foxi2 binding and gene expression in the early gastrula. Similarly, they perform ChIP-seq to map Sox3 binding and identify many similar trends as well as co-occupied Foxi2/Sox3 sites. Loss of function studies via MO knockdown demonstrates decrease in ectodermal gene expression in foxi2/sox3 double morphant. Intriguingly they also perform single-cell gene profiling, which reveals the specific genes and cell fates linked to co-binding of Foxi2/Sox3 in early gastrulation. Finally the authors demonstrate the sufficiency of Foxi2/Sox3 master transcription factors to delineate ectodermal gene expression. They ectopically express these factors in the endoderm, finding that only by co-expression do they observe a significant increase in ectoderm gene expression, along with now inappropriate p300 binding and H3K27ac at ectodermal genes within endoderm tissues. Thus analogous to the master regulator MyoD, these high level factors sufficient to push pluripotent cells into a specific cell fate, elevating their position withing the gene regulatory hierarchy governing one of the earliest differentiation events in development.

Strengths of the work include a tour-de-force of experimentation as well as epigenome data analyses, and clear and significant phenotypes from manipulating the master regulators Foxi2/Sox3, which demonstrate they are both necessary and sufficient to orchestrate the program of ectodermal gene expression. I have only a few comments and modest concerns, highlighted below

1. Further resolving dynamics of Foxi2/Sox3 binding, p300 recruitment and gene expression timing. The authors nicely demonstrate that class V-VII genes show increases or constant Foxi2/Sox3 binding during blastula and gastrula and robust transcription. Whereas Class I-III show diminished binding into gastrulation and little gene expression. However, they also claim Foxi2 and Sox3 also pre-bind CRMs for a variety of ectoderm genes at Stage 6.5 (using Chip-qPCR), which is many hours before any of their ChIP-seq data in later blastula. In constast they state that p300 and H3K27ac are not present at this stage. However, as far I can tell they do not provide and ChIP-QPCR data for these factors. It seems possible p300 may be partially enriched at proximal to some genes prior to genome activation in the mid-blastula. Overall I am left wondering the extent to which (1) these maternal factors are pre-bound to germ layer genes prior to genome activation - how many genes, what level of occupancy (compared to in later blastula), (2) whether this early binding is needed for later recruitment of p300, or (3) whether there may be low level recruitment of both the transcription factors and p300, which steadily increase over time during blastula and into gastrulation. In short, is there is a punctuated event, for example after genome activation, and how clearly does binding of TFs precede p300 and histone acetylation. I think the manuscript could be improved by clarifying these points

2. Moving supplemental figures into main text. I believe Supp Fig 3 is quite important and I kept referring back to it while reading the paper. This co-binding analysis for Foxi2 and Sox3 feels critical to understanding the manuscript and thus should probably be a main figure (new Figure 4?)

Rev. 2:

In the manuscript "Foxi2 and Sox3 are master regulators controlling ectoderm germ layer specification" Hendrickson and colleagues use Xenopus tropicalis embryos and explants to investigate how two maternally deposited transcription factors, Foxi2 and Sox3, confer ectoderm identity in the early blastula/gastrula embryo. Both Foxi2 and Sox3 were previously identified as regulators of early ectoderm development, however, how these factors regulate ectoderm induction remained unexplored, although, as the authors nicely discuss in their study, maternal deposition of these transcription factors likely extends to the entire amphibian and fish lineages. This indicates an evolutionarily relevant strategy to setup ectoderm identity in vertebrates. Using ChIP-seq on ectoderm vs. endoderm explants, they characterize differential histone modifications, Ep300 binding as well as Foxi2 and Sox3 bound regions in the embryo. By intersecting these data with transcriptional data on epidermal gene expression, they find that in the ectoderm, activating histone marks are associated with ectodermal expressed genes, while repressive marks are found close to endodermal expressed genes. Interestingly, analogous repressive marks on ectodermal genes are missing in the endoderm. Active regulatory regions, identified by overlap between Ep300 and H3K27ac, show strong enrichment for germ layer specific transcription factor motifs, and together with the animal maternal deposition of foxi2 and sox3 transcripts in the Xenopus egg, this indicates that Foxi2 and Sox3 broadly regulate ectodermal gene activation. Foxi2 and Sox3 ChIP-seq at different developmental stages further reveals that both transcription factors already bind to a subset of regulatory regions before genomic activation, suggesting that they could recruit Ep300 to the regulatory regions to promote robust ectodermal gene activation. Using ChIP-qPCR and single nucleus RNA-seq of gastrula stage embryos, they further validate a set of genes including ones regulated by both factors as well as by Foxi2 or Sox3 separately, and connect these genes to lineages in the early developing ectoderm (such as epidermis vs neuroectoderm or epithelial vs deep/generative layer of the ectoderm). Finally, they overexpress Foxi2 and Sox3 in the prospective endoderm and show that this leads to ectodermal gene expression as well as changes in epigenetic state of the endoderm, which takes on an ectodermal identity, including deposition of repressive marks at endodermal expressed genes. They further analyze large regulatory regions (ca. 20kb/locus) and argue that these represent super enhancers for the ectodermal lineage, which can be identified by co-binding of Foxi2, Sox3 and Ep300 (FSE regions). The pre-activation of these super enhancers is associated with reduced expression variance, i.e. the authors conclude that transcriptional noise is reduced and that this confers robustness in ectoderm specification. In the discussion, they highlight evolutionary relationships as well as propose a model in which early Foxi2 and Sox3 functions are being handed over to more sub-lineage specific transcription factors, such as Foxi1 (epidermis) and soxE family members (neuroectoderm).

Overall, this is a beautiful study employing a wide range of experimental techniques and assays to broaden our understanding of the step wise setup of differential developing regions in the vertebrate embryo. These insights are broadly relevant and with the specific factors and processes delineated in this study, further studies will be able to shed light on evolutionary adaptations in ectoderm specification as well as how cell lineages are regulated across tissues in the Xenopus embryo. Therefore, the study should be accepted for publication in PLOS Biology after some issues are clarified (see specific comments and questions below).

Main comments:

1.) Are most of the regions described in this paper really enhancers or rather promoters given the close proximity to the regulated genes? Perhaps it would be good to state specifically how the authors make that distinction and how they define the term here. Related to this: Is it correct to call Ep300 a marker of enhancers? Or just accessible chromatin?

2.) Why does it make sense that repressive marks on ectodermal genes are missing in endoderm? This is quite interesting, but not much discussed by the authors.

3.) Fig 3 A/B: class II and IV peaks are co occupied by Foxi2 and Sox3, but are excluded from analysis in Fig3D because low number of uniquely bound genes. This is confusing to me. Please clarify.

4.) Fig 4: Was this a mild phenotype or why does it make sense that ectodermal structures are still generated after MO against foxi2 and sox3? In sox3 morphants only 5 of 20 are having the phenotype. This makes me wonder about the robustness of the phenotype and why the embryos is shown as repressive?

5.) One of the main implicated consequences of their model is that Foxi2 and Sox3 are required for Ep300 to bind specific regions in the ectoderm to modify histones required for active gene expression. However, a definitive experiment addressing that is missing. Perhaps adding an experiment where Ep300 ChIP is performed after Foxi2 and Sox3 knockdown in ectoderm vs endoderm explants would be quite elucidating here. Ep300 binding should be lost in the ectoderm, but not in the endoderm.

Minor comments:

1.) Fig 1 SC color code needs clarification. No size of region indicated.

2.) Fig 2E: there is no rational given as to why it could make sense that foxi2 peaks coincide with sox/fox motifs but not fox/pou motifs. Would be nice to discuss/explain a bit more.

3.) Clarification on ChIP analysis: please state in the text that antibodies were used against endogenous proteins/markers and how these were validated to act specific. Also, the methods state that chip peaks were detected using input data from 2017. So no input data was generated with the presented Foxi2, Sox3, histone and Ep300 ChIP? This seems a bit odd. How do the authors make sure that technical artifacts are the same across studies/methods/reagents?

4.) It would be nice to discuss the hand over to other TFs and how that might affect sub-lineages in the ectoderm a bit more, however only when space permits.

---

## [Decision Letter · Decision Letter 2]

8 Sep 2025

Dear Dr Cho,

Thank you for your patience while we considered your revised manuscript entitled "Foxi2 and Sox3 are master regulators controlling ectoderm germ layer specification" for publication as a Research Article at PLOS Biology. This revised version of your manuscript has been evaluated by the PLOS Biology editors, the Academic Editor and one of the original reviewers.

Based on the review, we are likely to accept this manuscript for publication, provided you satisfactorily address the data and other policy-related requests stated below my signature.

In addition, we would like you to consider a suggestion to improve the title:

"Foxi2 and Sox3 are master transcription regulators that control ectoderm germ layer specification in Xenopus"

We expect to receive your revised manuscript within two weeks.

*Published Peer Review History*

*Press*

Sincerely,

Ines

--

Ines Alvarez-Garcia, PhD

Senior Editor

PLOS Biology

Fig. 1A, C; Fig. 2B-G; Fig. 3B-E; Fig. 4C-E; Fig. 5B; Fig. 6A-E, G-I; Fig. S2A, B; Fig. S3A, C; Fig. S4A, B; Fig. S5A and Fig. S6B-G

**Please also make the data you have deposited in GEO publicly available at this stage.

CODE POLICY

Reviewers' comment:

Rev. 2:

The authors have adequately addressed my previous comments, including by new experiements, and from my point of view the manuscript can be accepted for publication.

---

## [Editor Report · Decision Letter 3]

20 Oct 2025

Dear Dr Cho,

Thank you for the submission of your revised Research Article entitled "Foxi2 and Sox3 are master transcription regulators that control ectoderm germ layer specification in Xenopus" for publication in PLOS Biology. On behalf of my colleagues and the Academic Editor, Marianne Bronner, I am delighted to let you know that we can in principle accept your manuscript for publication, provided you address any remaining formatting and reporting issues. These will be detailed in an email you should receive within 2-3 business days from our colleagues in the journal operations team; no action is required from you until then. Please note that we will not be able to formally accept your manuscript and schedule it for publication until you have completed any requested changes.

PRESS

Sincerely, 

Ines

--

Ines Alvarez-Garcia, PhD,

Senior Editor

PLOS Biology
